# A Dual-Agent Adversarial Framework for Generalizable Reinforcement Learning

## Abstract

Recently, empowered with the powerful capabilities of neural networks, reinforcement learning (RL) has successfully tackled numerous challenging tasks. However, while these models demonstrate enhanced decision-making abilities, they are increasingly prone to overfitting. For instance, a trained RL model often fails to generalize to even minor variations of the same task, such as a change in background color or other minor semantic differences. To address this issue, we propose a dual-agent adversarial policy learning framework, which allows agents to spontaneously learn the underlying semantics without introducing any human prior knowledge. Specifically, our framework involves a game process between two agents: each agent seeks to maximize the impact of perturbing on the opponent's policy by producing representation differences for the same state, while maintaining its own stability against such perturbations. This interaction encourages agents to learn generalizable policies, capable of handling irrelevant features from the high-dimensional observations. Extensive experimental results on the Procgen benchmark demonstrate that the adversarial process significantly improves the generalization performance of both agents, while also being applied to various RL algorithms, e.g., Proximal Policy Optimization (PPO). With the adversarial framework, the RL agent outperforms the baseline methods by a significant margin, especially in hard-level tasks, marking a significant step forward in the generalization capabilities of deep reinforcement learning.

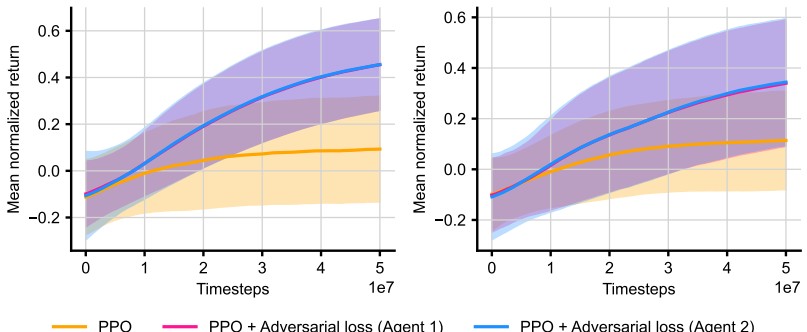

Figure 1: **Comparisons of the generalization capability of different RL agents.** By applying our adversarial approach to PPO, the two adversarial PPO agents demonstrate significant improvements in train performance (left) and test performance (right) across eight environments in the *hard-level* Procgen benchmark. In this context, higher scores indicate better generalization capabilities.

## 1 Introduction

Reinforcement Learning (RL) has emerged as a powerful paradigm for solving complex decision-making problems, leveraging an agent's ability to learn from interactions with an environment through trial and error (Sutton, 2018). However, generalization between tasks remains difficult for state-of-the-art deep reinforcement learning algorithms. Although trained agents can solve complex tasks, they often struggle to transfer their experience to new environments. For instance, an agent

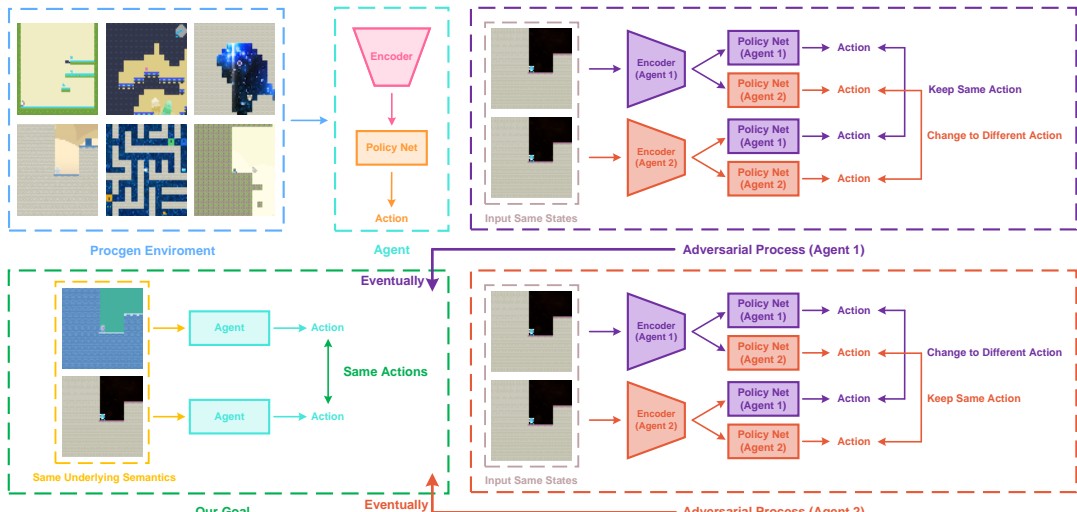

Figure 2: **Overview of the adversarial process.** Our method involves a game process between two homogeneous agents, as shown in the figure. The training samples are simultaneously input into the encoders of both agents, resulting in differing representations for the same observation. By adjusting the parameters of the two encoders, both agents aim to ensure that their own policy networks are robust to such differences while maximizing the influence of these differences on the other agent's policy network as much as possible. This minimax game process will eventually allow robust policy learning, preventing agents from overfitting to irrelevant features in high-dimensional observations, thereby enhancing generalization performance.

trained in a specific environment struggles to perform effectively in another, even when the only difference between environments is a subtle alteration, such as the change of colors in the scene (Cobbe et al., 2019; 2020). This limitation underscores the challenges of transferring knowledge across different contexts, emphasizing the importance of developing robust generalization strategies for RL applications in dynamic and variable real-world scenarios (Korkmaz, 2024).

One approach to enhancing generalization in RL focuses on data augmentation techniques (Lee et al., 2019; Laskin et al., 2020; Zhang & Guo, 2021), which increase the diversity of training data by modifying input observations or environmental conditions. While this provides a straightforward solution, it can introduce biases that do not align with RL objectives and often neglect the nuances of the RL process, potentially limiting effectiveness. Another approach involves regularizing the learned functions, drawing from traditional techniques used in deep neural networks, such as batch normalization (Liu et al., 2019), contrastive learning (Agarwal et al., 2021), and loss function regularization (Amit et al., 2020). However, these methods can not adequately address the unique challenges of RL, as they often focus on static representations rather than the dynamic nature of agent-environment interactions. Consequently, both data augmentation and traditional regularization methods have limitations that hinder their ability to facilitate effective generalization in RL.

Adversarial learning (Pinto et al., 2017; Zhang et al., 2020; Oikarinen et al., 2021; Li et al., 2021; Rahman & Xue, 2023) presents a promising direction for enhancing generalization in RL by learning robust representations of irrelevant features through an adversarial process. This framework facilitates the development of agents capable of adapting to new environments by emphasizing the distinction between relevant and irrelevant information. While adversarial learning frameworks integrate the RL process, existing methods often rely on introducing generator and discriminator networks (Goodfellow et al., 2014) or seek to modify fundamental parameters of the simulation environments. Such heterogeneous adversarial processes introduce additional hyperparameters and training costs, necessitating carefully designed architectures. These complexities make it challenging to establish a unified framework for generalization tasks across diverse domains.

To address the generalization problem in RL, in this paper, we propose a novel adversarial learning framework, which involves a game process between two homogeneous agents (in Figure 2). This framework offers three key advantages: 1). First, this general framework can integrate well

with existing policy learning algorithms such as Proximal Policy Optimization (PPO) (Schulman et al., 2017). 2). Second, the adversarial process allows agents to spontaneously learn the underlying semantics without necessitating additional human prior knowledge, thus fostering robust generalization performance. 3). Lastly, our approach introduces minimal additional hyperparameters, highlighting its potential for widespread applicability across various RL models. Extensive experiments demonstrate that our adversarial framework significantly improves generalization performance in Procgen (Cobbe et al., 2020), particularly in hard-level environments (in Figure 1). This framework marks a significant advancement in addressing generalization challenges in deep reinforcement learning.

Our contributions are summarized as follows:

- To the best of our knowledge, we are the first to theoretically prove that minimizing the policy's robustness to irrelevant features helps improve generalization performance.
- We propose a general adversarial learning framework to improve the generalization performance of agents, which is compatible with existing policy learning algorithms.
- Extensive results demonstrate that applying the adversarial framework to standard RL baselines gains significant improvements in generalization performance.

## 2    PRELIMINARIES

**Markov Decision Process and Generalization Settings.** We first consider the formalization of generalization in RL. Denote a Markov Decision Process (MDP) as $m$, defined by the tuple

$$m = (\mathcal{S}_m, \mathcal{A}, r_m, \mathcal{P}_m, \rho_m, \gamma), \tag{1}$$

where $m$ is sampled from the distribution $p_{\mathcal{M}}(\cdot)$, $\mathcal{S}_m$ represents the state space, $\mathcal{A}$ represents the action space, $r_m : \mathcal{S}_m \times \mathcal{A} \mapsto \mathbb{R}$ is the reward function, $\mathcal{P}_m : \mathcal{S}_m \times \mathcal{A} \times \mathcal{S}_m \mapsto [0, 1]$ is the probability distribution of the state transition function, $\rho_m : \mathcal{S}_m \mapsto [0, 1]$ is the probability distribution of the initial state, and $\gamma \in (0, 1]$ is the discount factor. Typically, during training, the agent is only allowed to access $\mathcal{M}_{\text{train}} \subset \mathcal{M}$ and is then tested for its generalization performance by extending to the entire distribution $\mathcal{M}$. The agent generates the following trajectory on $m$:

$$\tau_m = (s_0^m, a_0^m, r_0^m, \ldots, s_t^m, a_t^m, r_t^m, \ldots). \tag{2}$$

Similar to standard RL, the state-value function, value function can be defined as

$$Q_m^\pi(s_t^m, a_t^m) = \mathbb{E}_{s_{t+1}^m, a_{t+1}^m, \ldots} \left[ \sum_{k=0}^\infty \gamma^k r_m(s_{t+k}^m, a_{t+k}^m) \right], \quad V_m^\pi(s_t^m) = \mathbb{E}_{a_t^m \sim \pi(\cdot|s_t^m)} \left[ Q_m^\pi(s_t^m, a_t^m) \right]. \tag{3}$$

Given $Q_m^\pi$ and $V_m^\pi$, the advantage function can be expressed as $A_m^\pi(s_t^m, a_t^m) = Q_m^\pi(s_t^m, a_t^m) - V_m^\pi(s_t^m)$. We now denote $\zeta(\pi) = \mathbb{E}_{m \sim p_{\mathcal{M}}(\cdot), \tau_m \sim \pi} \left[ \sum_{t=0}^\infty \gamma^t r_m(s_t^m, a_t^m) \right]$ as the generalization objective given policy $\pi$, and denote $\eta(\pi) = \mathbb{E}_{m \sim p_{\mathcal{M}_{\text{train}}}(\cdot), \tau_m \sim \pi} \left[ \sum_{t=0}^\infty \gamma^t r_m(s_t^m, a_t^m) \right]$ as the training objective, where the notation $\mathbb{E}_{\tau_m \sim \pi}$ indicates the expected return of the trajectory $\tau_m$ generated by the agent following policy $\pi$, i.e., $s_0^m \sim \rho_m(\cdot), a_t^m \sim \pi(\cdot|s_t^m), r_t^m \sim r_m(s_t^m, a_t^m), s_{t+1}^m \sim \mathcal{P}_m(\cdot|s_t^m, a_t^m)$, where $t \in \mathbb{N}$, $\mathbb{N}$ is the set of all natural numbers.

For the convenience of subsequent theoretical analysis, we decouple the state $s_t^m$ into $u_t$ and $\phi_m(\cdot)$, i.e., $s_t^m = \phi_m(u_t)$, where $u_t$ is independent of $m$, while $\phi_m(\cdot)$ is completely and only determined by $m$. For instance, $u_t$ implicitly encompasses significant semantic information, which is crucial for the agent to maximize the expected return. This includes, for example, the relative positional relationship between the manipulated character and obstacles in its surroundings. On the other hand, the function $\phi_m$ obfuscates these pieces of information, such as the background or rendering style of the game. This suggests that even two vastly different states may represent identical semantics, making it seemingly implausible for an agent utilizing a Convolutional Neural Network (CNN) for feature extraction to maintain robustness against such variations.

Therefore, the generalization of reinforcement learning has been proven to be highly challenging (Ghosh et al., 2021), as the agent may use the additional information provided by $\phi_m$ to "cheat". In some extreme cases, the agent can achieve high scores solely by memorizing these additional pieces

of information, while lacking any comprehension of the underlying semantics, which can further lead to the agent completely failing on unseen $m \sim p_{\mathcal{M}}(\cdot)$.

Hence, we attempt to eliminate the influence of $m$. We consider a Hidden Markov Decision Process (HMDP) that consists entirely of useful information (in other words, all variables that can be affected by $m$ are excluded from consideration), denote it as $m^* = (\mathcal{U}, \mathcal{A}, r, \mathcal{P}, \rho, \gamma)$.

## 3 THEORETICAL ANALYSIS

In this section, we derive the lower bounds for the training and generalization performance of the agent. The main conclusion drawn from this is that improving the agent's robustness to irrelevant features will help enhance its generalization performance.

Given the probability distribution $p_{\mathcal{M}}$, we first make the following assumption:

**Assumption 3.1.** When $m$ is sampled from $\mathcal{M}_{\text{train}} \subset \mathcal{M}$, i.e., $m \sim p_{\mathcal{M}_{\text{train}}}(\cdot)$, we assume that

$$p_{\mathcal{M}_{\text{train}}}(m) = \frac{p_{\mathcal{M}}(m) \cdot \mathbb{I}(m \in \mathcal{M}_{\text{train}})}{M}, \tag{4}$$

where $M = \int_{\mathcal{M}_{\text{train}}} p_{\mathcal{M}}(m) \mathrm{d}m$ is the normalized coefficient that represents the probability that $m$, sampled from the entire distribution $\mathcal{M}$, belongs to $\mathcal{M}_{\text{train}}$, while $\mathbb{I}(\cdot)$ is the indicator function.

It can be proved that $p_{\mathcal{M}_{\text{train}}}(m)$ is a probability distribution, please refer to Appendix C.1 for details. Based on Assumption 3.1, we can derive the following generalization theorem:

**Theorem 3.2** (Generalization performance lower bound). *Given any policy $\pi$, the following bound holds:*

$$\zeta(\pi) \geq \eta(\pi) - \frac{2r_{\max}}{1 - \gamma} \cdot (1 - M), \tag{5}$$

*where $\zeta(\pi)$ and $\eta(\pi)$ denote the generalization objective and training objective, respectively; $r_{\max} = \max_{m,s,a} |r_m(s,a)|$.*

The proof is in Appendix C.2. This inspires us that when sampling $m$ from the entire $\mathcal{M}$, with the increase of $M$ (i.e., the probability of the sampled $m \in \mathcal{M}_{\text{train}}$), the lower bound of generalization performance is continuously optimized and tends to be consistent with $\zeta$ when $M = 1$.

According to Theorem 3.2, once $\mathcal{M}_{\text{train}}$ is determined, the value of $M$ is also fixed, at this point, $\eta$ is the only term that we can optimize in the lower bound. Therefore, we now focus on optimizing $\eta$. Before that, we present some important theoretical results in the following:

**Theorem 3.3.** *(Kakade & Langford, 2002) Let $\mathbb{P}(s_t = s|\pi)$ represents the probability of the $t$-th state equals to $s$ in trajectories generated by the agent following policy $\pi$, and $\rho_\pi(s) = \sum_{t=0}^{\infty} \gamma^t \mathbb{P}(s_t = s|\pi)$ represents the unnormalized discounted visitation frequencies. Given any two policies, $\pi$ and $\tilde{\pi}$, their performance difference can be measured by*

$$\eta(\tilde{\pi}) = \eta(\pi) + \mathbb{E}_{s \sim \rho_{\tilde{\pi}}(\cdot), a \sim \tilde{\pi}(\cdot|s)} \left[ A^\pi(s,a) \right]. \tag{6}$$

**Theorem 3.4.** *(Schulman, 2015) Given any two policies, $\pi$ and $\tilde{\pi}$, the following bound holds:*

$$\eta(\tilde{\pi}) \geq L_\pi(\tilde{\pi}) - \frac{4\gamma \max_{s,a} |A^\pi(s,a)|}{(1 - \gamma)^2} \cdot D_{\text{TV}}^{\max}(\pi, \tilde{\pi})^2, \tag{7}$$

*where $L_\pi(\tilde{\pi}) = \eta(\pi) + \mathbb{E}_{s \sim \rho_\pi(\cdot), a \sim \tilde{\pi}(\cdot|s)} \left[ A^\pi(s,a) \right]$.*

The aforementioned theorems only consider standard RL. On this foundation, we further extend them and derive a lower bound for the training objective:

**Theorem 3.5** (Training performance lower bound). *Let $\mathbb{P}(s_t^m = s|m, \pi)$ represents the probability of the $t$-th state equals to $s$ in trajectories generated by the agent following policy $\pi$ in MDP $m$, and $\rho_\pi^m(s) = \sum_{t=0}^{\infty} \gamma^t \mathbb{P}(s_t^m = s|m, \pi)$ represents the unnormalized discounted visitation frequencies. Given any two policies, $\pi$ and $\tilde{\pi}$, the following bound holds:*

$$\eta(\tilde{\pi}) \geq L_\pi(\tilde{\pi}) - \frac{4\gamma A_{\max}}{(1 - \gamma)^2} \cdot \left( \sqrt{\mathfrak{D}_1} + \sqrt{\mathfrak{D}_2} + \sqrt{\mathfrak{D}_3} \right)^2, \tag{8}$$

---

**Algorithm 1** Policy iteration algorithm guaranteeing non-decreasing training performance $\eta$

---

1: **Initialize:** policy $\pi_0$
2: **for** $i = 0, 1, 2, \ldots$ **do**
3:     Solve the constrained optimization problem through

$$\pi_{i+1} \leftarrow \arg\max_{\pi} \ L_{\pi_i}(\pi) - \eta(\pi_i) - M_{\pi_i}(\pi)$$

$$\text{s.t.} \quad L_{\pi_i}(\pi) - \eta(\pi_i) \geq M_{\pi_i}(\pi)$$

4: **end for**

---

*where $A_{\max} = \max_{m,s,a} |A_m^\pi(s,a)|$, and*

$$\eta(\tilde{\pi}) = \eta(\pi) + \mathbb{E}_{m \sim p_{\mathcal{M}_{\text{train}}}(\cdot), s \sim \rho_{\tilde{\pi}}^m(\cdot), a \sim \tilde{\pi}(\cdot|s)} \left[ A_m^\pi(s,a) \right],$$

$$L_\pi(\tilde{\pi}) = \eta(\pi) + \mathbb{E}_{m \sim p_{\mathcal{M}_{\text{train}}}(\cdot), s \sim \rho_\pi^m(\cdot), a \sim \tilde{\pi}(\cdot|s)} \left[ A_m^\pi(s,a) \right],$$

$$\mathfrak{D}_1 = \mathbb{E}_{m \sim p_{\mathcal{M}_{\text{train}}}(\cdot)} \left\{ D_{\text{TV}}^{\max} \left[ \pi(\cdot|\phi_m(u)), \tilde{\pi}(\cdot|\phi_m(u)) \right]^2 \right\}, \tag{9}$$

$$\mathfrak{D}_2 = \mathbb{E}_{m, \tilde{m} \sim p_{\mathcal{M}_{\text{train}}}(\cdot)} \left\{ D_{\text{TV}}^{\max} \left[ \pi(\cdot|\phi_m(u)), \pi(\cdot|\phi_{\tilde{m}}(u)) \right]^2 \right\},$$

$$\mathfrak{D}_3 = \mathbb{E}_{m, \tilde{m} \sim p_{\mathcal{M}_{\text{train}}}(\cdot)} \left\{ D_{\text{TV}}^{\max} \left[ \tilde{\pi}(\cdot|\phi_m(u)), \tilde{\pi}(\cdot|\phi_{\tilde{m}}(u)) \right]^2 \right\},$$

*where the notation $D_{\text{TV}}^{\max}(\cdot) = \max_u D_{\text{TV}}(\cdot)$.*

The proof see Appendix C.3. This inspires us that $\mathfrak{D}_1$ measures the difference between the old and new policies, while $\mathfrak{D}_2$ and $\mathfrak{D}_3$ represent the robustness of the old and new policies to irrelevant features of the high-dimensional observations, respectively. Thus

$$\eta(\tilde{\pi}) - \eta(\pi) \geq L_\pi(\tilde{\pi}) - \eta(\pi) - C \cdot \left( \sqrt{\mathfrak{D}_1} + \sqrt{\mathfrak{D}_2} + \sqrt{\mathfrak{D}_3} \right)^2, \tag{10}$$

where $C = 4\gamma A_{\max}/(1-\gamma)^2$. We now denote $M_\pi(\tilde{\pi}) = C \cdot \left( \sqrt{\mathfrak{D}_1} + \sqrt{\mathfrak{D}_2} + \sqrt{\mathfrak{D}_3} \right)^2$, we can then derive the following monotonic improvement theorem:

**Theorem 3.6** (Monotonic improvement of training performance). *Let $\pi_0, \pi_1, \pi_2, \ldots, \pi_k$ be the sequence of policies generated by Algorithm 1, then*

$$\eta(\pi_k) \geq \eta(\pi_{k-1}) \geq \cdots \geq \eta(\pi_0). \tag{11}$$

*Proof.* According to inequality (10) and Algorithm 1, we have

$$\eta(\pi_{i+1}) - \eta(\pi_i) \geq L_{\pi_i}(\pi_{i+1}) - \eta(\pi_i) - M_{\pi_i}(\pi_{i+1}) \geq 0, \tag{12}$$

where $i = 0, 1, \ldots, k-1$, so that $\eta(\pi_{i+1}) \geq \eta(\pi_i)$, concluding the proof. $\square$

On the other hand, it is evident that

$$\eta(\pi_{i+1}) - \frac{2r_{\max}}{1-\gamma} \cdot (1 - M) \geq \eta(\pi_i) - \frac{2r_{\max}}{1-\gamma} \cdot (1 - M), \tag{13}$$

which means through the iterative process of Algorithm 1, we optimize the lower bound of generalization performance (5) as well. In fact, if both $\mathfrak{D}_2$ and $\mathfrak{D}_3$ are always equal to zero, i.e., given any $m, \tilde{m} \in \mathcal{M}$ and $u \in \mathcal{U}$, we have $\pi_i(\cdot|\phi_m(u)) = \pi_i(\cdot|\phi_{\tilde{m}}(u)), \forall i \in \mathbb{N}$. In this case, the agent has complete insight into the underlying semantics without any influence from $m$, thus the agent is essentially interacting with $m^* = (\mathcal{U}, \mathcal{A}, r, \mathcal{P}, \rho, \gamma)$, Theorem 3.5 degenerates into Theorem 3.4.

However, Algorithm 1 is an idealized approach, we have to adopt some heuristic approximations in practical solutions. In the following section, we will discuss the specific details of these approximations and introduce our proposed dual-agent adversarial framework to overcome the difficulty in optimizing the lower bound (10), which constitutes the core of this paper.

# 4 METHODOLOGY

In the previous section, we derived the lower bound of training performance, which inspires us to optimize the part of the policy that determines robustness. Therefore, in this section, we first analyze the optimization problem of parameterized policies (Section 4.1), then deconstruct what properties a generalization agent should have (Section 4.2), and finally propose a dual-agent adversarial framework to solve the generalization problem (Section 4.3).

## 4.1 OPTIMIZATION OF PARAMETERIZED POLICIES

We first consider the parameterized policies, i.e., $\pi_\theta$, and denote the upstream encoder of the policy network as $\psi_w$, where $w$ and $\theta$ represent the parameters of the encoder and policy network, respectively.

For any given state $s = \phi_m(u)$, for brevity, we denote $\bar{s}_m = \psi_w(\phi_m(u))$ as the representation input into the policy network $\pi_\theta$ after passing through the encoder $\psi_w$. Similar to TRPO (Schulman, 2015), the total variational distance and KL divergence satisfy $D_{\mathrm{TV}}^{\max}\left[\pi_{\theta_{\mathrm{old}}}(\cdot|\bar{s}_m), \pi_\theta(\cdot|\bar{s}_m)\right]^2 \leq D_{\mathrm{KL}}^{\max}\left[\pi_{\theta_{\mathrm{old}}}(\cdot|\bar{s}_m), \pi_\theta(\cdot|\bar{s}_m)\right]$, where $\theta_{\mathrm{old}}$ represents the policy network parameters before the update, while $\theta$ represents the current policy network parameters. Through heuristic approximation, the maximum KL divergence $D_{\mathrm{KL}}^{\max}$ is approximated as the average KL divergence $\mathbb{E}\left[D_{\mathrm{KL}}\right]$, and then Algorithm 1 is approximated as the following constrained optimization problem:

$$\max_\theta \ J(\theta) = L_{\theta_{\mathrm{old}}}(\theta) - \eta(\theta_{\mathrm{old}}),$$

$$\text{s.t.} \begin{cases} \mathbb{E}_{m \sim p_{\mathcal{M}_{\mathrm{train}}}(\cdot)}\left\{D_{\mathrm{KL}}\left[\pi_{\theta_{\mathrm{old}}}(\cdot|\bar{s}_m), \pi_\theta(\cdot|\bar{s}_m)\right]\right\} \leq \delta_1, \\ \mathbb{E}_{m,\tilde{m} \sim p_{\mathcal{M}_{\mathrm{train}}}(\cdot)}\left\{D_{\mathrm{KL}}\left[\pi_\theta(\cdot|\bar{s}_m), \pi_\theta(\cdot|\bar{s}_{\tilde{m}})\right]\right\} \leq \delta_2, \end{cases} \tag{14}$$

where $m$ and $\tilde{m}$ are MDPs independently sampled from the distribution $p_{\mathcal{M}_{\mathrm{train}}}$. Then, similar to TRPO, $J(\theta)$ can be expressed as

$$J(\theta) = \mathbb{E}_{m,s;a \sim \pi_\theta(\cdot|\bar{s}_m)}\left[A_m^\pi(s,a)\right] = \mathbb{E}_{m,s;a \sim \pi_{\theta_{\mathrm{old}}}(\cdot|\bar{s}_m)}\left[\frac{\pi_\theta(a|\bar{s}_m)}{\pi_{\theta_{\mathrm{old}}}(a|\bar{s}_m)} \cdot A_m^\pi(s,a)\right], \tag{15}$$

which is called importance sampling, where $m \sim p_{\mathcal{M}_{\mathrm{train}}}(\cdot)$ and $s \sim \rho_{\pi_{\theta_{\mathrm{old}}}}^m(\cdot)$. Thus, we can further transform the constrained optimization problem (14) into the following form:

$$\max_\theta \ J(\theta) = \mathbb{E}_{m,s;a \sim \pi_{\theta_{\mathrm{old}}}(\cdot|\bar{s}_m)}\left[\frac{\pi_\theta(a|\bar{s}_m)}{\pi_{\theta_{\mathrm{old}}}(a|\bar{s}_m)} \cdot \hat{A}(s,a)\right],$$

$$\text{s.t.} \begin{cases} \mathbb{E}_{m \sim p_{\mathcal{M}_{\mathrm{train}}}(\cdot)}\left\{D_{\mathrm{KL}}\left[\pi_{\theta_{\mathrm{old}}}(\cdot|\bar{s}_m), \pi_\theta(\cdot|\bar{s}_m)\right]\right\} \leq \delta_1, \\ \mathbb{E}_{m,\tilde{m} \sim p_{\mathcal{M}_{\mathrm{train}}}(\cdot)}\left\{D_{\mathrm{KL}}\left[\pi_\theta(\cdot|\bar{s}_m), \pi_\theta(\cdot|\bar{s}_{\tilde{m}})\right]\right\} \leq \delta_2, \end{cases} \tag{16}$$

where $\hat{A}(s,a)$ is the estimation of the advantage function, and in this paper, we adopt the GAE (Schulman et al., 2015) technique. The first constraint of (16) measures the difference between the old and new policies, where TRPO (Schulman, 2015) and PPO (Schulman et al., 2017) have already provided corresponding solutions. However, it's important to note that the second constraint in (16) can not be approximated, as it involves different states with the same underlying semantics, and predicting another $\phi_{\tilde{m}}(u)$ based on any received state $\phi_m(u)$ ($m \neq \tilde{m}$) is untraceble.

Hence, understanding different states with the same underlying semantics is the most central challenge in the generalization of deep reinforcement learning. In the following section, we will systematically discuss the characteristics a sufficiently general agent should possess to achieve good generalization performance.

## 4.2 HOW TO ACHIEVE GOOD GENERALIZATION?

As discussed previously, it is unable to solve the optimization problem (16) directly, as the expectation $\mathbb{E}_{m,\tilde{m} \sim p_{\mathcal{M}_{\mathrm{train}}}(\cdot)}\left\{D_{\mathrm{KL}}\left[\pi_\theta(\cdot|\bar{s}_m), \pi_\theta(\cdot|\bar{s}_{\tilde{m}})\right]\right\}$ cannot be estimated due to the unknown distribution $p_{\mathcal{M}_{\mathrm{train}}}$ and function $\phi_m$. In this section, we focus on analyzing the characteristics that a generalization agent should possess.

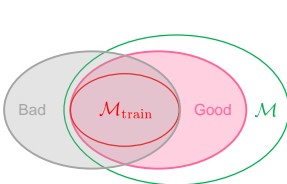
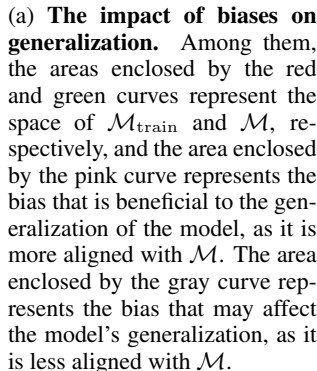

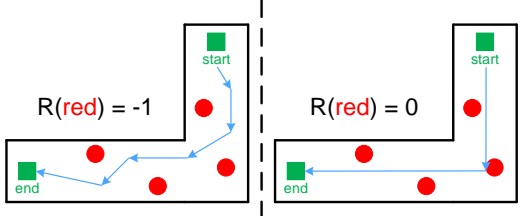

(a) **The impact of biases on generalization.** Among them, the areas enclosed by the red and green curves represent the space of $\mathcal{M}_{\text{train}}$ and $\mathcal{M}$, respectively, and the area enclosed by the pink curve represents the bias that is beneficial to the generalization of the model, as it is more aligned with $\mathcal{M}$. The area enclosed by the gray curve represents the bias that may affect the model's generalization, as it is less aligned with $\mathcal{M}$.

(b) **The impact of reward functions on generalization.** We build a simple maze environment where an agent represented by a green square starts from the starting point and can receive a reward by reaching the endpoint. However, there is a possibility that the agent may enter the red zone. The reward functions are set as follows: the agent receives a reward of $-1$ after entering the red zone (left), and the agent receives a reward of $0$ after entering the red zone (right). It is evident that in the left environment, the positional information of the red zone is useful to the agent, while in the right environment, the positional information of the red zone can be ignored by the agent. Thus, the agent should have different representations for the two environments.

Figure 3: The impacts of biases and reward functions on generalization

Although estimating different states with the same semantics during training is challenging, one effective approach to explicitly learn the underlying semantics is to introduce the adversarial method. For instance, Rahman & Xue (2023) aims to maximize expected return while minimizing the interference from adversarial examples generated by its own generator, StarGAN (Choi et al., 2018). This process ultimately facilitates robust policy learning and helps prevent the agent from overfitting irrelevant features in high-dimensional observations, inspiring us to incorporate an adversarial framework into our approach (Section 4.3).

However, StarGAN does not entirely eliminate the biases introduced by human prior knowledge. Specifically, the domain of the original input image is clustered using a Gaussian Mixture Model (GMM), which inherently introduces biases from the GMM. Furthermore, the number of clusters is often determined empirically, adding another layer of human influence.

Therefore, firstly, a sufficiently general agent should spontaneously learn robust representations for irrelevant features, rather than relying on biases introduced by human prior knowledge. Figure 3 (a) shows the potential impact of introducing biases into the model. Secondly, the entire pipeline for learning generalization must integrate the RL process, as the identification of irrelevant features is closely linked to the objectives of RL, particularly the configuration of the reward function. Figure 3 (b) demonstrates how different reward functions influence the agent's recognition of irrelevant information within a simple maze environment.

In summary, we conclude that a sufficiently general agent should possess two characteristics:

**(1) The agent is able to spontaneously learn robust representations for high-dimensional input without introducing any bias that benefits from human prior knowledge.**

**(2) The agent should adaptively adjust its representation of underlying semantics in response to changes in the reward function, demonstrating the ability to identify the semantics corresponding to specific objectives.**

Given these two points, we will introduce a dual-agent adversarial framework in the following section, which empowers agents with enhanced generalization capabilities.

### 4.3 ADVERSARIAL POLICY LEARNING FRAMEWORK

In the previous analysis, we summarized the core challenges in the generalization of RL (Section 4.1) and the characteristics a general agent should possess (Section 4.2). However, generating adversarial

---

**Algorithm 2** Dual-agent adversarial policy learning

---

1: **Initialize:** Agent 1's encoder and policy $\psi_1, \pi_1$, agent 2's encoder and policy $\psi_2, \pi_2$
2: **Initialize:** Reinforcement learning algorithm $\mathcal{A}$
3: **Initialize:** Gradient descent optimizer $\mathcal{O}$
4: **while** training **do**
5:    **for** $i = 1, 2$ **do**
6:       Collect data $\mathcal{D}_i$ using agent $i$
7:       Calculate RL loss for agent $i$: $\mathcal{L}_{\mathrm{RL}} \leftarrow \mathcal{A}(\mathcal{D}_i)$
8:       Calculate KL loss for agent $i$: $\mathcal{L}_{\mathrm{KL}} \leftarrow D_{\mathrm{KL}}^{\mathrm{own}} - D_{\mathrm{KL}}^{\mathrm{other}}$ according to Equation (19)
9:       Calculate total loss for agent $i$: $\mathcal{L} \leftarrow \mathcal{L}_{\mathrm{RL}} + \alpha \mathcal{L}_{\mathrm{KL}}$
10:      Update $\psi_i, \pi_i, \psi_{3-i} \leftarrow \mathcal{O}(\mathcal{L}, \psi_i, \pi_i, \psi_{3-i})$
11:    **end for**
12: **end while**

---

samples through generative models introduces additional hyperparameters and training costs, and relies on carefully designed model architecture.

To address these issues, a viable solution is to attack the agent's encoder instead of directly generating adversarial samples. In this section, we introduce a dual-agent adversarial framework, which involves a game process between two homogeneous agents, as shown in Figure 4.

In particular, two symmetric agents are introduced in this framework, both agents have the capability to utilize their respective training data and update the other agent's encoder through backpropagation, which empowers them to perform adversarial attacks on each other. Since the two agents are equivalent in status, we take the perspective of agent 1 as an example. Agent 1 inputs its training data $s_1$ into both its own encoder and the other agent's encoder, obtaining $\psi_1(s_1)$ and $\psi_2(s_1)$, resulting in different representations of the same state $s_1$. The adversarial framework consists of two processes:

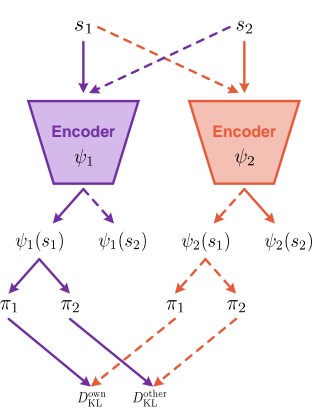

**Adversarial Attack on Opponent Agent.** To prevent the opponent agent from producing good actions, agent 1 attempts to alter the parameters of both encoders to influence agent 2's decision-making, where the KL divergence is used to quantify this distributional perturbation:

$$D_{\mathrm{KL}}^{\mathrm{other}} = D_{\mathrm{KL}}\left[\pi_2(\cdot|\psi_2(s_1)), \pi_2(\cdot|\psi_1(s_1))\right]. \quad (17)$$

**Robust Defense Against Adversarial Threats.** Meanwhile, agent 1 itself attempts to remain robust to this influence, which can be expressed as

$$D_{\mathrm{KL}}^{\mathrm{own}} = D_{\mathrm{KL}}\left[\pi_1(\cdot|\psi_1(s_1)), \pi_1(\cdot|\psi_2(s_1))\right]. \quad (18)$$

It should be noted that when agent 1 is performing adversarial attacks on agent 2's encoder $\psi_2$, the parameters of agent 2's policy network $\pi_2$ are frozen during this stage, thus do not participate in gradient updates.

Overall, the goal of the agent is to maximize the perturbation $D_{\mathrm{KL}}^{\mathrm{other}}$ while minimizing the self-inference $D_{\mathrm{KL}}^{\mathrm{own}}$, resulting in the loss function of the form:

Figure 4: **Adversarial policy learning framework.** $\psi_1$ and $\pi_1$ represent the encoder and policy network of agent 1, while $\psi_2$ and $\pi_2$ represent the encoder and policy network of agent 2. $s_1$ and $s_2$ represent the training data for agent 1 and agent 2, respectively. Moreover, the solid lines indicate that the training data of each agent is input into its corresponding encoder, while dashed lines indicate that the training data of each agent is input into the other's encoder.

$$\mathcal{L}_{\mathrm{KL}} = D_{\mathrm{KL}}^{\mathrm{own}} - D_{\mathrm{KL}}^{\mathrm{other}}. \quad (19)$$

Since the adversarial process is coupled with the RL training process, the total loss is defined as

$$\mathcal{L} = \mathcal{L}_{\mathrm{RL}} + \alpha \mathcal{L}_{\mathrm{KL}}, \quad (20)$$

where $\mathcal{L}_{\mathrm{RL}}$ is the loss function using a specific RL algorithm, $\alpha$ is the only additional hyperparameter. As the two agents are equivalent, the training processes for both agents are completely symmetrical. The pseudo-code of the adversarial policy learning process is shown in Algorithm 2.

The overall loss comprises two components: the reinforcement learning loss term $\mathcal{L}_{\text{RL}}$ and the adversarial loss term $\mathcal{L}_{\text{KL}}$. The adversarial loss facilitates a competitive interaction among agents and functions similarly to a form of regularization, effectively preventing agents from overfitting to irrelevant features in high-dimensional observations. With the alternate updating of the two agents, they will have to consider the truly useful underlying semantics, leading to better generalization performance, or mathematically speaking, a lower $\mathbb{E}_{m,\tilde{m}\sim p_{\mathcal{M}_{\text{train}}}(\cdot)}\left\{ D_{\text{KL}}\left[\pi_\theta(\cdot|\bar{s}_m), \pi_\theta(\cdot|\bar{s}_{\tilde{m}})\right]\right\}$ in constrained optimization problem (16).

In summary, our proposed adversarial policy learning framework is well in line with the two characteristics proposed in Section 4.2:

**(1) First, the framework does not introduce any additional biases, allowing the agents to learn the underlying semantics spontaneously.**

**(2) Second, the adversarial process and the reinforcement learning process are highly coupled, which means that the dependency between the reward signal and the corresponding representation can be modeled well.**

## 5 EXPERIMENTS

### 5.1 EXPERIMENTAL SETTINGS

**Benchmark.** Procgen (Cobbe et al., 2020) is an environment library specifically designed for reinforcement learning research, developed by OpenAI. It provides a diverse and procedurally generated set of platform games, allowing researchers to test the generalization capabilities of agents across different tasks and scenarios.

**Baselines.** We verify the performance of our proposed method compared with PPO (Schulman et al., 2017) and DAAC (Raileanu & Fergus, 2021) as the baselines for our comparative experiments.

**Training Settings.** In all experiments, we use the hyperparameters provided in the Appendix B unless otherwise specified. We referred to the original paper for hyperparameters specific to the algorithm. Following the recommendations of Cobbe et al. (2020), we run these methods on hard-level generalization tasks, training on eight environments of 500 levels and evaluating generalization performance on the full distribution of levels. We interact for 50M steps to consider running time. This is sufficient to assess the performance differences between our method and other baselines.

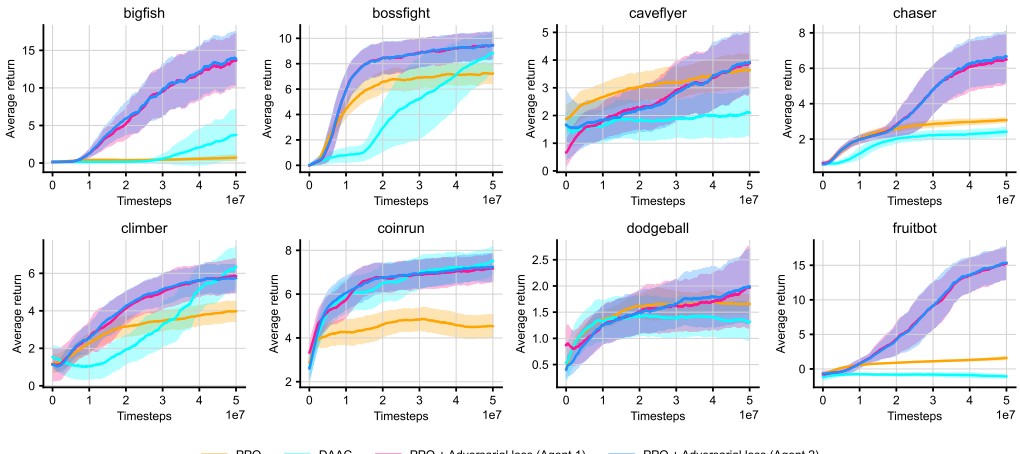

Figure 5: **Test performance curves of each method on eight hard-level Procgen games.** Each agent is trained on 500 training levels for 50M environment steps and evaluated on the full distribution of levels. The mean and standard deviation is shown across three random seeds.

Table 1: **Average test performance** of PPO, DAAC and PPO with our adversarial loss on eight hard-level Procgen games. The average return is shown across three random seeds.

| Env. \ Method | PPO | DAAC | PPO + Adv. (Agent 1) | PPO + Adv. (Agent 2) |
|---|---|---|---|---|
| bigfish | 0.485 | 1.193 | 7.904 | **8.117** |
| bossfight | 6.196 | 4.655 | 7.957 | **7.970** |
| caveflyer | **3.162** | 1.852 | 2.768 | 2.731 |
| chaser | 2.634 | 1.955 | 4.215 | **4.270** |
| climber | 3.233 | 3.299 | 4.473 | **4.520** |
| coinrun | 4.592 | 6.735 | 6.710 | **6.803** |
| dodgeball | 1.587 | 1.380 | 1.554 | **1.593** |
| fruitbot | 1.037 | -0.860 | 8.027 | **8.050** |
| *Average Score* | 2.866 | 2.526 | 5.451 | **5.507** |

## 5.2 EXPERIMENT RESULTS

Our experimental results are illustrated in Figure 5 and Table 1. The data clearly demonstrate that directly integrating our adversarial framework with the Proximal Policy Optimization (PPO) algorithm leads to substantial performance enhancements across various environments. In particular, our PPO + Adv. methods consistently outperform the DAAC algorithm, which relies on carefully crafted model architectures and additional hyperparameters. For instance, in the chaser task, our agent achieves a score of 4.270, surpassing DAAC's score of 1.955. This highlights not only the strength of our framework but also its potential to simplify model design while enhancing performance. In addition, the performance of our approach achieves an impressive score of 8.117 in bigfish, representing a remarkable increase of 1573% compared to the baseline PPO score of 0.485. Similarly, in the fruitbot environment, our method records a score of 8.050, a significant improvement of 676% over the PPO score of 1.037. These examples underscore the effectiveness of our adversarial approach in facilitating robust learning and adaptation in complex scenarios.

Overall, the average scores reveal that our methods yield an average score of 5.507, compared to 2.866 for the standard PPO and 2.526 for DAAC. This improvement reflects a strong generalization capability, indicating that our framework enables agents to perform better across a range of environments, thereby enhancing their adaptability and resilience to variations in task conditions.

## 6 RELATED WORK

**Generalizable RL Methods.** Data augmentation methods are considered as effective solutions for enhancing the generalization of agents. Directly integrating existing data augmentation methods with RL algorithms can yield improvements (Laskin et al., 2020; Kostrikov et al., 2020; Zhang & Guo, 2021; Raileanu et al., 2021). Domain randomization techniques (Tobin et al., 2017; Yue et al., 2019; Slaoui et al., 2019; Lee et al., 2019; Mehta et al., 2020; Li et al., 2021) inject random disturbances representing variations in the simulated environment during the training process of RL, effectively enhancing the adaptability of RL agents to unknown environments.

**Adversarial Learning.** Adversarial learning has been proven to be a powerful learning framework (Goodfellow et al., 2014; Jiang et al., 2020; Dong et al., 2020). For instance, combining adversarial learning with randomization to enhance the generalization performance of agents (Pinto et al., 2017; Li et al., 2021; Rahman & Xue, 2023). In addition, adversarial attacks are also used to improve the robustness and generalization performance of agents (Gleave et al., 2019; Oikarinen et al., 2021).

## 7 CONCLUSION

This paper introduces a dual-agent adversarial framework designed to tackle the challenges of generalization in reinforcement learning. By incorporating a competitive process between two agents, our framework leverages adversarial loss to enable both agents to spontaneously learn effective representations of high-dimensional observations, resulting in robust policies that effectively handle irrelevant features. Extensive experimental results demonstrate that this framework significantly enhances both the training and generalization performance of baseline RL algorithms. Our findings indicate that the adversarial approach not only improves the resilience of RL agents but also represents a meaningful advancement in the quest for generalizable reinforcement learning solutions.

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

## A  TRAINING RESULTS

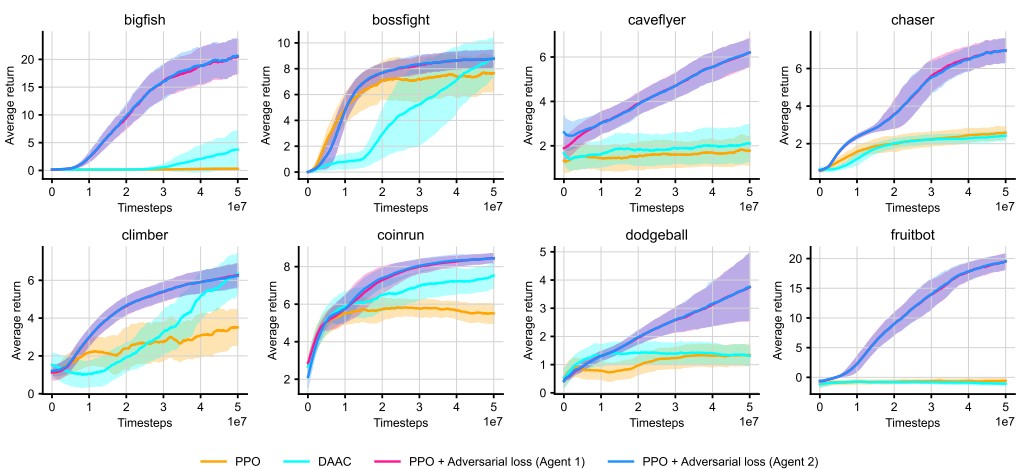

PPO    DAAC    PPO + Adversarial loss (Agent 1)    PPO + Adversarial loss (Agent 2)

Figure 6: **Train performance curves of each method on eight hard-level Procgen games.**

## B  HYPERPARAMETER SETTINGS

Table 2: Detailed hyperparameters in Procgen.

| Hyperparameters | PPO (Schulman et al., 2017) | DAAC (Raileanu & Fergus, 2021) | PPO with adversarial loss (ours) |
|---|---|---|---|
| Environments per worker | 64 | 64 | 64 |
| Workers | 4 | 4 | 4 |
| Horizon | 256 | 256 | 256 |
| Learning rate | $5 \times 10^{-4}$ | $5 \times 10^{-4}$ | $5 \times 10^{-4}$ |
| Learning rate decay | No | No | No |
| Optimizer | Adam | Adam | Adam |
| Total steps | 50M | 50M | 50M |
| Batch size | 16384 | 16384 | 16384 |
| Update epochs | 3 | - | 3 |
| Mini-batches | 8 | 8 | 8 |
| Mini-batch size | 2048 | 2048 | 2048 |
| GAE parameter $\lambda$ | 0.95 | 0.95 | 0.95 |
| Discount factor $\gamma$ | 0.999 | 0.999 | 0.999 |
| Value loss coefficient $c_1$ | 0.5 | - | 0.5 |
| Entropy loss coefficient $c_2$ | 0.01 | 0.01 | 0.01 |
| Probability ratio parameter $\epsilon$ | 0.2 | 0.2 | 0.2 |
| KL loss coefficient $\alpha$ | - | - | 1.0 |
| Advantage loss coefficient $\alpha_a$ | - | 0.25 | - |
| Policy update epochs $E_\pi$ | - | 1 | - |
| Value update epochs $E_V$ | - | 9 | - |
| Value updates after a policy update $N_\pi$ | - | 1 | - |

## C  PROOFS

### C.1  PROOF OF ASSUMPTION 3.1

We now prove that $p_{\mathcal{M}_{\mathrm{train}}}(m)$ is a probability distribution, by integrating it, we obtain

$$
\begin{aligned}
\int_{\mathcal{M}_{\mathrm{train}}} p_{\mathcal{M}_{\mathrm{train}}}(m)\mathrm{d}m &= \int_{\mathcal{M}_{\mathrm{train}}} \frac{p_{\mathcal{M}}(m) \cdot \mathbb{I}\left(m \in \mathcal{M}_{\mathrm{train}}\right)}{M}\mathrm{d}m \\
&= \frac{1}{M} \int_{\mathcal{M}_{\mathrm{train}}} p_{\mathcal{M}}(m) \cdot \mathbb{I}\left(m \in \mathcal{M}_{\mathrm{train}}\right)\mathrm{d}m \\
&= \frac{1}{M} \int_{\mathcal{M}_{\mathrm{train}}} p_{\mathcal{M}}(m)\mathrm{d}m \\
&= 1,
\end{aligned}
\tag{21}
$$

concluding the proof.

## C.2 PROOF OF THEOREM 3.2

We are trying to measure the difference between $\zeta(\pi)$ and $\eta(\pi)$, which is

$$
\begin{aligned}
& |\zeta(\pi) - \eta(\pi)| \\
&= \left| \mathbb{E}_{m \sim p_{\mathcal{M}}(\cdot), \tau_m \sim \pi} \left[ \sum_{t=0}^{\infty} \gamma^t r_m(s_t^m, a_t^m) \right] - \mathbb{E}_{m \sim p_{\mathcal{M}_{\text{train}}}(\cdot), \tau_m \sim \pi} \left[ \sum_{t=0}^{\infty} \gamma^t r_m(s_t^m, a_t^m) \right] \right| \\
&= \left| \mathbb{E}_{m \sim p_{\mathcal{M}}(\cdot)} \left\{ \mathbb{E}_{\tau_m \sim \pi} \left[ \sum_{t=0}^{\infty} \gamma^t r_m(s_t^m, a_t^m) \right] \right\} - \mathbb{E}_{m \sim p_{\mathcal{M}_{\text{train}}}(\cdot)} \left\{ \mathbb{E}_{\tau_m \sim \pi} \left[ \sum_{t=0}^{\infty} \gamma^t r_m(s_t^m, a_t^m) \right] \right\} \right|.
\end{aligned}
\tag{22}
$$

First, we denote $\mathbb{E}_{\tau_m \sim \pi} \left[ \sum_{t=0}^{\infty} \gamma^t r_m(s_t^m, a_t^m) \right]$ as $g_m(\pi)$, then

$$
\begin{aligned}
|g_m(\pi)| &= \left| \mathbb{E}_{\tau_m \sim \pi} \left[ \sum_{t=0}^{\infty} \gamma^t r_m(s_t^m, a_t^m) \right] \right| \\
&= \left| \sum_{t=0}^{\infty} \sum_s \mathbb{P}(s_t^m = s|m, \pi) \sum_a \pi(a|s) \cdot \gamma^t r_m(s, a) \right| \\
&= \left| \sum_{t=0}^{\infty} \gamma^t \sum_s \mathbb{P}(s_t^m = s|m, \pi) \sum_a \pi(a|s) \cdot r_m(s, a) \right| \\
&\leq \sum_{t=0}^{\infty} \gamma^t \cdot \max_{m,s,a} |r_m(s, a)| \\
&= \frac{r_{\max}}{1 - \gamma}.
\end{aligned}
\tag{23}
$$

Second, according to Assumption 3.1, we have

$$
\begin{aligned}
& |\zeta(\pi) - \eta(\pi)| \\
&= \left| \mathbb{E}_{m \sim p_{\mathcal{M}}(\cdot)} [g_m(\pi)] - \mathbb{E}_{m \sim p_{\mathcal{M}_{\text{train}}}(\cdot)} [g_m(\pi)] \right| \\
&= \left| \int_{\mathcal{M}} p_{\mathcal{M}}(m) g_m(\pi) \mathrm{d}m - \int_{\mathcal{M}_{\text{train}}} p_{\mathcal{M}_{\text{train}}}(m) g_m(\pi) \mathrm{d}m \right| \\
&= \left| \int_{\mathcal{M}_{\text{train}}} p_{\mathcal{M}}(m) g_m(\pi) \mathrm{d}m - \int_{\mathcal{M}_{\text{train}}} p_{\mathcal{M}_{\text{train}}}(m) g_m(\pi) \mathrm{d}m + \int_{\mathcal{M} - \mathcal{M}_{\text{train}}} p_{\mathcal{M}}(m) g_m(\pi) \mathrm{d}m \right| \\
&= \left| \left( 1 - \frac{1}{M} \right) \int_{\mathcal{M}_{\text{train}}} p_{\mathcal{M}}(m) g_m(\pi) \mathrm{d}m + \int_{\mathcal{M} - \mathcal{M}_{\text{train}}} p_{\mathcal{M}}(m) g_m(\pi) \mathrm{d}m \right| \\
&\leq \left| \left( 1 - \frac{1}{M} \right) \int_{\mathcal{M}_{\text{train}}} p_{\mathcal{M}}(m) g_m(\pi) \mathrm{d}m \right| + \left| \int_{\mathcal{M} - \mathcal{M}_{\text{train}}} p_{\mathcal{M}}(m) g_m(\pi) \mathrm{d}m \right| \\
&\leq \left( \frac{1}{M} - 1 \right) \cdot \frac{r_{\max}}{1 - \gamma} \cdot \int_{\mathcal{M}_{\text{train}}} p_{\mathcal{M}}(m) \mathrm{d}m + \frac{r_{\max}}{1 - \gamma} \cdot \int_{\mathcal{M} - \mathcal{M}_{\text{train}}} p_{\mathcal{M}}(m) \mathrm{d}m \\
&= \left( \frac{1}{M} - 1 \right) \cdot \frac{r_{\max}}{1 - \gamma} \cdot M + \frac{r_{\max}}{1 - \gamma} \cdot (1 - M) \\
&= \frac{2 r_{\max}}{1 - \gamma} \cdot (1 - M).
\end{aligned}
\tag{24}
$$

Theorem 3.2 follows.

## C.3 PROOF OF THEOREM 3.5

Let's start with Theorem 3.4 (Schulman, 2015), through a simple extension, by adding expectation $\mathbb{E}_{m \sim p_{\mathcal{M}_{\text{train}}(\cdot)}}$ to the left and right sides of Theorem 3.4, we can derive the following lemma:

**Lemma C.1.** *Let $m \sim p_{\mathcal{M}_{\text{train}}(\cdot)}$, given any two policies, $\pi$ and $\tilde{\pi}$, the following bound holds:*

$$\eta(\tilde{\pi}) \geq L_\pi(\tilde{\pi}) - \frac{4\gamma A_{\max}}{(1-\gamma)^2} \cdot \mathbb{E}_{m \sim p_{\mathcal{M}_{\text{train}}(\cdot)}} \left\{ D_{\text{TV}}^{\max} \left[ \pi(\cdot|\phi_m(u)), \tilde{\pi}(\cdot|\phi_m(u)) \right]^2 \right\}, \tag{25}$$

*where $A_{\max} = \max_{m,s,a} |A_m^\pi(s,a)|$ and*

$$\begin{aligned}
\eta(\tilde{\pi}) &= \eta(\pi) + \mathbb{E}_{m \sim p_{\mathcal{M}_{\text{train}}(\cdot), s \sim \rho_{\tilde{\pi}}^m(\cdot), a \sim \tilde{\pi}(\cdot|s)}} \left[ A_m^\pi(s,a) \right], \\
L_\pi(\tilde{\pi}) &= \eta(\pi) + \mathbb{E}_{m \sim p_{\mathcal{M}_{\text{train}}(\cdot), s \sim \rho_\pi^m(\cdot), a \sim \tilde{\pi}(\cdot|s)}} \left[ A_m^\pi(s,a) \right].
\end{aligned} \tag{26}$$

*Proof.* According to Theorem 3.4, given any $m$, we have

$$\begin{aligned}
&\left| \mathbb{E}_{s \sim \rho_{\tilde{\pi}}^m(\cdot), a \sim \tilde{\pi}(\cdot|s)} \left[ A_m^\pi(s,a) \right] - \mathbb{E}_{s \sim \rho_\pi^m(\cdot), a \sim \tilde{\pi}(\cdot|s)} \left[ A_m^\pi(s,a) \right] \right| \\
&\leq \frac{4\gamma \max_{s,a} |A_m^\pi(s,a)|}{(1-\gamma)^2} \cdot D_{\text{TV}}^{\max} \left[ \pi(\cdot|\phi_m(u)), \tilde{\pi}(\cdot|\phi_m(u)) \right]^2,
\end{aligned} \tag{27}$$

where the notation $D_{\text{TV}}^{\max}(\cdot) = \max_u D_{\text{TV}}(\cdot)$. Then

$$\begin{aligned}
&|\eta(\tilde{\pi}) - L_\pi(\tilde{\pi})| \\
&= \left| \mathbb{E}_{m \sim p_{\mathcal{M}_{\text{train}}(\cdot), s \sim \rho_{\tilde{\pi}}^m(\cdot), a \sim \tilde{\pi}(\cdot|s)}} \left[ A_m^\pi(s,a) \right] - \mathbb{E}_{m \sim p_{\mathcal{M}_{\text{train}}(\cdot), s \sim \rho_\pi^m(\cdot), a \sim \tilde{\pi}(\cdot|s)}} \left[ A_m^\pi(s,a) \right] \right| \\
&= \left| \mathbb{E}_{m \sim p_{\mathcal{M}_{\text{train}}(\cdot)}} \left\{ \mathbb{E}_{s \sim \rho_{\tilde{\pi}}^m(\cdot), a \sim \tilde{\pi}(\cdot|s)} \left[ A_m^\pi(s,a) \right] - \mathbb{E}_{s \sim \rho_\pi^m(\cdot), a \sim \tilde{\pi}(\cdot|s)} \left[ A_m^\pi(s,a) \right] \right\} \right| \\
&\leq \mathbb{E}_{m \sim p_{\mathcal{M}_{\text{train}}(\cdot)}} \left\{ \left| \mathbb{E}_{s \sim \rho_{\tilde{\pi}}^m(\cdot), a \sim \tilde{\pi}(\cdot|s)} \left[ A_m^\pi(s,a) \right] - \mathbb{E}_{s \sim \rho_\pi^m(\cdot), a \sim \tilde{\pi}(\cdot|s)} \left[ A_m^\pi(s,a) \right] \right| \right\} \\
&\leq \mathbb{E}_{m \sim p_{\mathcal{M}_{\text{train}}(\cdot)}} \left\{ \frac{4\gamma \max_{s,a} |A_m^\pi(s,a)|}{(1-\gamma)^2} \cdot D_{\text{TV}}^{\max} \left[ \pi(\cdot|\phi_m(u)), \tilde{\pi}(\cdot|\phi_m(u)) \right]^2 \right\} \\
&\leq \frac{4\gamma A_{\max}}{(1-\gamma)^2} \cdot \mathbb{E}_{m \sim p_{\mathcal{M}_{\text{train}}(\cdot)}} \left\{ D_{\text{TV}}^{\max} \left[ \pi(\cdot|\phi_m(u)), \tilde{\pi}(\cdot|\phi_m(u)) \right]^2 \right\},
\end{aligned} \tag{28}$$

Lemma C.1 follows. $\qquad\square$

Since the expectation of any constant is still this constant, i.e., $\mathbb{E}[c] = c$, we have

$$\begin{aligned}
&\mathbb{E}_{m \sim p_{\mathcal{M}_{\text{train}}(\cdot)}} \left\{ D_{\text{TV}}^{\max} \left[ \pi(\cdot|\phi_m(u)), \tilde{\pi}(\cdot|\phi_m(u)) \right]^2 \right\} \\
&= \mathbb{E}_{\tilde{m} \sim p_{\mathcal{M}_{\text{train}}(\cdot)}} \left\{ \mathbb{E}_{m \sim p_{\mathcal{M}_{\text{train}}(\cdot)}} \left\{ D_{\text{TV}}^{\max} \left[ \pi(\cdot|\phi_m(u)), \tilde{\pi}(\cdot|\phi_m(u)) \right]^2 \right\} \right\} \\
&= \mathbb{E}_{m,\tilde{m} \sim p_{\mathcal{M}_{\text{train}}(\cdot)}} \left\{ D_{\text{TV}}^{\max} \left[ \pi(\cdot|\phi_m(u)), \tilde{\pi}(\cdot|\phi_m(u)) \right]^2 \right\},
\end{aligned} \tag{29}$$

thus

$$\eta(\tilde{\pi}) \geq L_\pi(\tilde{\pi}) - \frac{4\gamma A_{\max}}{(1-\gamma)^2} \cdot \mathbb{E}_{m,\tilde{m} \sim p_{\mathcal{M}_{\text{train}}(\cdot)}} \left\{ D_{\text{TV}}^{\max} \left[ \pi(\cdot|\phi_m(u)), \tilde{\pi}(\cdot|\phi_m(u)) \right]^2 \right\}. \tag{30}$$

Now, denote $u^* = \arg\max_u D_{\text{TV}} \left[ \pi(\cdot|\phi_m(u)), \tilde{\pi}(\cdot|\phi_m(u)) \right]^2$, and based on the triangle inequality for total variation distance, we have

$$\begin{aligned}
&\mathbb{E}_{m,\tilde{m} \sim p_{\mathcal{M}_{\text{train}}(\cdot)}} \left\{ D_{\text{TV}}^{\max} \left[ \pi(\cdot|\phi_m(u)), \tilde{\pi}(\cdot|\phi_m(u)) \right]^2 \right\} \\
&= \mathbb{E}_{m,\tilde{m} \sim p_{\mathcal{M}_{\text{train}}(\cdot)}} \left\{ D_{\text{TV}} \left[ \pi(\cdot|\phi_m(u^*)), \tilde{\pi}(\cdot|\phi_m(u^*)) \right]^2 \right\} \\
&\leq \mathbb{E}_{m,\tilde{m} \sim p_{\mathcal{M}_{\text{train}}(\cdot)}} \Big\{ \big( D_{\text{TV}} \left[ \pi(\cdot|\phi_m(u^*)), \pi(\cdot|\phi_{\tilde{m}}(u^*)) \right] + D_{\text{TV}} \left[ \pi(\cdot|\phi_{\tilde{m}}(u^*)), \tilde{\pi}(\cdot|\phi_{\tilde{m}}(u^*)) \right] + \\
&\quad + D_{\text{TV}} \left[ \tilde{\pi}(\cdot|\phi_m(u^*)), \tilde{\pi}(\cdot|\phi_{\tilde{m}}(u^*)) \right] \big)^2 \Big\},
\end{aligned} \tag{31}$$

so that

$$
\begin{aligned}
&\mathbb{E}_{m,\tilde{m}\sim p_{\mathcal{M}_{\text{train}}}(\cdot)}\left\{ D_{\text{TV}}^{\max}\left[\pi(\cdot|\phi_m(u)),\tilde{\pi}(\cdot|\phi_m(u))\right]^2 \right\}\\
&\leq \mathbb{E}_{m,\tilde{m}\sim p_{\mathcal{M}_{\text{train}}}(\cdot)}\left\{ D_{\text{TV}}\left[\pi(\cdot|\phi_m(u^*)),\pi(\cdot|\phi_{\tilde{m}}(u^*))\right]^2 \right\}\\
&+\mathbb{E}_{m,\tilde{m}\sim p_{\mathcal{M}_{\text{train}}}(\cdot)}\left\{ D_{\text{TV}}\left[\pi(\cdot|\phi_{\tilde{m}}(u^*)),\tilde{\pi}(\cdot|\phi_{\tilde{m}}(u^*))\right]^2 \right\}\\
&+\mathbb{E}_{m,\tilde{m}\sim p_{\mathcal{M}_{\text{train}}}(\cdot)}\left\{ D_{\text{TV}}\left[\tilde{\pi}(\cdot|\phi_m(u^*)),\tilde{\pi}(\cdot|\phi_{\tilde{m}}(u^*))\right]^2 \right\}\\
&+2\mathbb{E}_{m,\tilde{m}\sim p_{\mathcal{M}_{\text{train}}}(\cdot)}\left\{ D_{\text{TV}}\left[\pi(\cdot|\phi_m(u^*)),\pi(\cdot|\phi_{\tilde{m}}(u^*))\right]\cdot D_{\text{TV}}\left[\pi(\cdot|\phi_{\tilde{m}}(u^*)),\tilde{\pi}(\cdot|\phi_{\tilde{m}}(u^*))\right] \right\}\\
&+2\mathbb{E}_{m,\tilde{m}\sim p_{\mathcal{M}_{\text{train}}}(\cdot)}\left\{ D_{\text{TV}}\left[\pi(\cdot|\phi_m(u^*)),\pi(\cdot|\phi_{\tilde{m}}(u^*))\right]\cdot D_{\text{TV}}\left[\tilde{\pi}(\cdot|\phi_m(u^*)),\tilde{\pi}(\cdot|\phi_{\tilde{m}}(u^*))\right] \right\}\\
&+2\mathbb{E}_{m,\tilde{m}\sim p_{\mathcal{M}_{\text{train}}}(\cdot)}\left\{ D_{\text{TV}}\left[\pi(\cdot|\phi_{\tilde{m}}(u^*)),\tilde{\pi}(\cdot|\phi_{\tilde{m}}(u^*))\right]\cdot D_{\text{TV}}\left[\tilde{\pi}(\cdot|\phi_m(u^*)),\tilde{\pi}(\cdot|\phi_{\tilde{m}}(u^*))\right] \right\}.
\end{aligned}
\tag{32}
$$

Next, according to the Cauchy-Schwarz inequality, i.e., $X$ and $Y$ are two positive random variables, then $\mathbb{E}\left[XY\right]\leq\sqrt{\mathbb{E}\left[X^2\right]\cdot\mathbb{E}\left[Y^2\right]}$, we obtain

$$
\begin{aligned}
&\mathbb{E}_{m,\tilde{m}\sim p_{\mathcal{M}_{\text{train}}}(\cdot)}\left\{ D_{\text{TV}}^{\max}\left[\pi(\cdot|\phi_m(u)),\tilde{\pi}(\cdot|\phi_m(u))\right]^2 \right\}\\
&\leq \mathbb{E}_{m,\tilde{m}\sim p_{\mathcal{M}_{\text{train}}}(\cdot)}\left\{ D_{\text{TV}}\left[\pi(\cdot|\phi_m(u^*)),\pi(\cdot|\phi_{\tilde{m}}(u^*))\right]^2 \right\}\\
&+\mathbb{E}_{m,\tilde{m}\sim p_{\mathcal{M}_{\text{train}}}(\cdot)}\left\{ D_{\text{TV}}\left[\pi(\cdot|\phi_{\tilde{m}}(u^*)),\tilde{\pi}(\cdot|\phi_{\tilde{m}}(u^*))\right]^2 \right\}\\
&+\mathbb{E}_{m,\tilde{m}\sim p_{\mathcal{M}_{\text{train}}}(\cdot)}\left\{ D_{\text{TV}}\left[\tilde{\pi}(\cdot|\phi_m(u^*)),\tilde{\pi}(\cdot|\phi_{\tilde{m}}(u^*))\right]^2 \right\}\\
&+2\sqrt{\mathbb{E}_{m,\tilde{m}\sim p_{\mathcal{M}_{\text{train}}}(\cdot)}\left\{ D_{\text{TV}}\left[\pi(\cdot|\phi_m(u^*)),\pi(\cdot|\phi_{\tilde{m}}(u^*))\right]^2 \right\}\cdot\mathbb{E}_{m,\tilde{m}\sim p_{\mathcal{M}_{\text{train}}}(\cdot)}\left\{ D_{\text{TV}}\left[\pi(\cdot|\phi_{\tilde{m}}(u^*)),\tilde{\pi}(\cdot|\phi_{\tilde{m}}(u^*))\right]^2 \right\}}\\
&+2\sqrt{\mathbb{E}_{m,\tilde{m}\sim p_{\mathcal{M}_{\text{train}}}(\cdot)}\left\{ D_{\text{TV}}\left[\pi(\cdot|\phi_m(u^*)),\pi(\cdot|\phi_{\tilde{m}}(u^*))\right]^2 \right\}\cdot\mathbb{E}_{m,\tilde{m}\sim p_{\mathcal{M}_{\text{train}}}(\cdot)}\left\{ D_{\text{TV}}\left[\tilde{\pi}(\cdot|\phi_m(u^*)),\tilde{\pi}(\cdot|\phi_{\tilde{m}}(u^*))\right]^2 \right\}}\\
&+2\sqrt{\mathbb{E}_{m,\tilde{m}\sim p_{\mathcal{M}_{\text{train}}}(\cdot)}\left\{ D_{\text{TV}}\left[\pi(\cdot|\phi_{\tilde{m}}(u^*)),\tilde{\pi}(\cdot|\phi_{\tilde{m}}(u^*))\right]^2 \right\}\cdot\mathbb{E}_{m,\tilde{m}\sim p_{\mathcal{M}_{\text{train}}}(\cdot)}\left\{ D_{\text{TV}}\left[\tilde{\pi}(\cdot|\phi_m(u^*)),\tilde{\pi}(\cdot|\phi_{\tilde{m}}(u^*))\right]^2 \right\}}\\
&\leq \mathfrak{D}_2+\mathfrak{D}_1+\mathfrak{D}_3+2\sqrt{\mathfrak{D}_2\mathfrak{D}_1}+2\sqrt{\mathfrak{D}_2\mathfrak{D}_3}+2\sqrt{\mathfrak{D}_1\mathfrak{D}_3}\\
&=\left(\sqrt{\mathfrak{D}_1}+\sqrt{\mathfrak{D}_2}+\sqrt{\mathfrak{D}_3}\right)^2.
\end{aligned}
\tag{33}
$$

Finally, by combining the inequality (30) and inequality (33), we derive

$$
\eta(\tilde{\pi})\geq L_\pi(\tilde{\pi})-\frac{4\gamma A_{\max}}{(1-\gamma)^2}\cdot\left(\sqrt{\mathfrak{D}_1}+\sqrt{\mathfrak{D}_2}+\sqrt{\mathfrak{D}_3}\right)^2,
\tag{34}
$$

concluding the proof of Theorem 3.5.

