# OpenReview forum: "A Dual-Agent Adversarial Framework for Generalizable Reinforcement Learning"
_ICLR.cc/2025/Conference — ICLR 2025 Conference Withdrawn Submission_

### Official Review · Reviewer_Pk9A · 2024-10-24

**Soundness:** 3
**Presentation:** 3
**Contribution:** 3
**Rating:** 5
**Confidence:** 3

**Summary:**

The paper presents a dual-agent adversarial framework aimed at improving the generalization capabilities of reinforcement learning (RL) models, which often struggle with overfitting and fail to adapt to minor variations in tasks. The proposed framework facilitates a game process between two agents that learn to perturb each other’s policies while maintaining their own stability, enabling them to focus on relevant features in high-dimensional observations. Extensive experiments on the Procgen benchmark demonstrate that this adversarial approach significantly enhances the agents’ performance, especially in challenging environments, outperforming traditional RL algorithms like Proximal Policy Optimization (PPO). Additionally, the authors theoretically prove that reducing an agent’s robustness to irrelevant features can improve its generalization performance. Overall, the study marks a significant advancement in addressing generalization challenges in deep reinforcement learning.

**Strengths:**

- The introduction of a dual-agent adversarial framework is an innovative approach that addresses the pressing issue of overfitting and generalization in reinforcement learning, offering a new perspective on how agents can improve adaptability in varying environments.
- The paper provides a strong theoretical foundation, proving that reducing an agent’s robustness to irrelevant features can lead to better generalization, enhancing the depth of the contribution.
- The experiments conducted on the Procgen benchmark show significant performance improvements over existing methods like PPO, demonstrating the effectiveness of the proposed framework in real-world, challenging tasks.
- By focusing on reducing overfitting and enhancing generalization, the paper addresses a critical gap in reinforcement learning research, providing solutions applicable to broader, more complex environments.
- The framework is well-designed to scale across different environments, making it applicable to a wide range of RL tasks with high-dimensional observations.

**Weaknesses:**

- While the framework performs well on the Procgen benchmark, its applicability to real-world tasks remains untested, leaving questions about how well it generalizes outside controlled environments.
- The dual-agent adversarial framework introduces additional computational complexity, which may pose challenges in terms of scalability and efficiency for resource-constrained systems.
- Although the framework is shown to improve generalization, more detailed ablation studies could have been included to clarify the contribution of individual components, such as the specific impact of the adversarial training mechanism.
- The paper assumes that irrelevant features can be identified and suppressed, but it does not sufficiently address how to detect these features in environments where their classification is unclear or context-dependent.
- The comparison with state-of-the-art methods is somewhat limited, with a stronger focus on performance gains rather than in-depth analysis of differences in behavior between approaches.

**Questions:**

- How does the adversarial training framework handle environments where the distinction between relevant and irrelevant features is not well-defined or context-dependent?
- What strategies could be employed to reduce the computational overhead introduced by the dual-agent setup, especially in more complex or resource-constrained environments?
- Could the method be extended or adapted to improve generalization in real-world tasks beyond the Procgen benchmark, and what modifications would be necessary to achieve this?
- What impact would the framework have in environments with continuous action spaces or higher-dimensional state representations, where irrelevant features may be harder to isolate?
- How would performance vary in scenarios where adversarial training results in catastrophic forgetting of useful features, and what mechanisms could prevent this?
- Are there any considerations for applying this approach to tasks with dynamic or evolving feature relevance, where the set of relevant features may change over time?

---

> ### Author Response · Authors · 2024-11-28
>
> Dear Reviewer Pk9A,
>
> Thank you for your strong support of our work. We will now address your concerns.
>
> >**While the framework performs well on the Procgen benchmark, its applicability to real-world tasks remains untested, leaving questions about how well it generalizes outside controlled environments.**
>
> >**Could the method be extended or adapted to improve generalization in real-world tasks beyond the Procgen benchmark, and what modifications would be necessary to achieve this?**
>
> Thank you for your thoughtful suggestion. However, Procgen is one of the most widely used benchmarks for testing the generalization ability of agents. Typically, testing generalization performance on Procgen is sufficient to reflect the quality of an algorithm. We have noted that many high-quality published works have also used Procgen as their primary testing benchmark [1, 2, 3, 4, 5].
>
> >**The dual-agent adversarial framework introduces additional computational complexity, which may pose challenges in terms of scalability and efficiency for resource-constrained systems.**
>
> We acknowledge that our framework does introduce additional computational complexity. However, in terms of the performance improvements achieved, the added computational complexity is acceptable. Additionally, we would like to emphasize the theoretical contributions of this work.
>
> >**Although the framework is shown to improve generalization, more detailed ablation studies could have been included to clarify the contribution of individual components, such as the specific impact of the adversarial training mechanism.**
>
> Thank you very much for your insightful suggestion. Our framework appears to be plug-and-play, so we would like to gently remind you that, apart from combining our algorithm with the baseline PPO algorithm, there do not seem to be additional ablation experiments. The experimental results also indicate that our framework significantly improves the performance of the baseline algorithm (Figures 5 and 6).
>
> >**The paper assumes that irrelevant features can be identified and suppressed, but it does not sufficiently address how to detect these features in environments where their classification is unclear or context-dependent.**
>
> >**How does the adversarial training framework handle environments where the distinction between relevant and irrelevant features is not well-defined or context-dependent?**
>
> Thank you for your suggestion. However, these irrelevant features are learned spontaneously by the agent and are not influenced by factors such as the contextual environment.
>
> >**The comparison with state-of-the-art methods is somewhat limited, with a stronger focus on performance gains rather than in-depth analysis of differences in behavior between approaches.**
>
> We do not entirely agree with your point. We indeed emphasize performance improvement (through combining our method with the baseline algorithm) rather than comparing our performance against SOTA algorithms. Additionally, we have thoroughly analyzed why our method can enhance generalization performance (Sections 3 and 4). We also provide very detailed theoretical results (Section 3), which is one of the contributions that cannot be overlooked.

---

> ### Author Response · Authors · 2024-11-28
>
> >**What strategies could be employed to reduce the computational overhead introduced by the dual-agent setup, especially in more complex or resource-constrained environments?**
>
> That's a great question! We have also attempted to reduce computational overhead, but at this time, we are unable to provide a better solution. We are very eager to explore potential ways to reduce computational costs in the future. Thank you again for your suggestion.
>
> >**What impact would the framework have in environments with continuous action spaces or higher-dimensional state representations, where irrelevant features may be harder to isolate?**
>
> Since the Procgen environment uses purely visual inputs, we believe that high-dimensional state representations have already been proven effective. Additionally, we acknowledge the current lack of further exploration into continuous action spaces. We would be delighted to explore these possibilities in future work!
>
> However, despite this, our theoretical results are not affected by changes in the state space or action space, as all our theoretical results rely only on mild assumptions.
>
> >**How would performance vary in scenarios where adversarial training results in catastrophic forgetting of useful features, and what mechanisms could prevent this?**
>
> >**Are there any considerations for applying this approach to tasks with dynamic or evolving feature relevance, where the set of relevant features may change over time?**
>
> Thank you for your question! However, the Procgen environment is static. Our study is limited to this static environment rather than environments where agents need to continuously learn, which would require avoiding the forgetting of useful features.
>
>
> ## References
>
> [1] K Cobbe et al. Leveraging procedural generation to benchmark reinforcement learning.
>
> [2] K Cobbe et al. Quantifying generalization in reinforcement learning.
>
> [3] R Raileanu et al. Decoupling value and policy for generalization in reinforcement learning.
>
> [4] R Raileanu et al. Automatic data augmentation for generalization in deep reinforcement learning.
>
> [5] A Jesson et al. Improving Generalization on the ProcGen Benchmark with Simple Architectural Changes and Scale.

---

### Official Review · Reviewer_GnKk · 2024-11-02

**Soundness:** 1
**Presentation:** 2
**Contribution:** 1
**Rating:** 3
**Confidence:** 4

**Summary:**

The submission proposes an adversarial framework that involves a game process between two agents: each agent seeks to maximize the impact of perturbing the opponent’s policy by producing representation differences for the same state, while maintaining its own stability against such perturbations. The submission conducts experiments in the ProcGen environment with 3 random seeds in 8 different games and provides comparison against DAAC and PPO.

**Strengths:**

Generalization in deep reinforcement learning is a highly important research direction.

**Weaknesses:**

The theoretical claims of the submission follow almost immediately from previous work, and do not bring any additional new knowledge.

Table 1 should include standard deviations. Three random seeds is relatively small to interpret the results reported. The results reported in Table 1 and Figure 5 are contradictory. Table 1 reports that DAAC in climber is 3.299 and PPO + Adv. (Agent 1) is 4.473. However, Figure 5 clearly reports that DAAC performance as the highest. How is this possible?

Why is it only compared to DAAC and original PPO? There are more studies on generalization in deep reinforcement learning.

In the DAAC paper there is another algorithm called IDAAC that performs better. Why is the algorithm IDAAC not included in the comparison?

I would also recommend checking page 9 and page 10 of the ProcGen section of paper [1]. In particular, the paper [1] states for ProcGen that:

*“We note that a number of improvements reported in the existing literature are only 50 − 70% likely.”*

Furthermore the paper [1] states:

*“Instead, we recommend using normalization based on the estimated minimum and maximum scores on ProcGen and reporting aggregate metrics based on such score.”*

As it has been reported in [1] and [3], the performance of PPG [2] is also quite high. It might be good to include PPG in the comparison baseline.

ProcGen seems to have 16 tasks. Both of these papers [1,2] test across the 16 games in the ProcGen environment. The submission tests their proposed algorithm in only 8 of them.

[1] Deep Reinforcement Learning at the Edge of the Statistical Precipice, NeurIPS 2021.

[2] Phasic Policy Gradient, ICML 2021.

[3] Decoupling Value and Policy for Generalization in Reinforcement Learning, ICML 2021.

More recent techniques report substantially higher scores in the ProcGen environment [1,2].

[1] DRIBO: Robust Deep Reinforcement Learning via Multi-View Information Bottleneck, ICML 2022.

[2] Explore to Generalize in Zero-Shot RL, NeurIPS 2023.


How adversarial learning is mentioned in the introduction is incorrect. In the introduction it is stated that:

*“Adversarial framework facilitates the development of agents capable of adapting to new environments by emphasizing the distinction between relevant and irrelevant information.”*

by referring to these studies [1,2,3] as adversarial learning

[1] State-Adversarial DQN for robust deep reinforcement learning, NeurIPS 2020.

[2] Robust adversarial reinforcement learning, ICML 2017.

[3] Robust Deep Reinforcement Learning through Adversarial Loss, NeurIPS 2021.

However, recent studies demonstrated that in fact adversarially trained policies cannot generalize, and furthermore the generalization skills of standard reinforcement learning training is substantially higher [1].

[1] Adversarial Robust Deep Reinforcement Learning Requires Redefining Robustness, AAAI 2023.


Since the submission proposes an adversarial training method it would have been good to test against adversarial examples as well. It does not have to be the most state-of-the-art adversarial attacks, but still it would have been good to include for reference.


Another thing I want to mention is that by employing the proposed adversarial learning framework the number of encoder parameters that needs to be trained is in fact doubled. This brings a new set of questions. Is it really a fair comparison to lower capacity models as previous ones? Would the prior methods perform also well if we simply just increased the parameters in the encoder?

**Questions:**

Please see above.

---

> ### Author Response · Authors · 2024-11-28
>
> Dear Reviewer GnKk,
>
> Thank you for your constructive feedback. We will address your concerns as follows:
>
> >**The theoretical claims of the submission follow almost immediately from previous work, and do not bring any additional new knowledge.**
>
> Thank you for your feedback. However, we respectfully disagree with your viewpoint. Firstly, our theoretical results are primarily derived from the mild assumption 3.1, which we have not seen in previous works. Secondly, we derived the generalization performance lower bound (Theorem 3.2) and the training performance lower bound (Theorem 3.5) based on assumption 3.1. Although the derivation of Theorem 3.5 is partly based on previous results, we believe this conclusion is non-trivial, revealing some profound aspects of reinforcement learning generalization. Specifically, it shows that enhancing the agent's robustness to irrelevant features can improve generalization, which aligns well with our intuition.
>
> >**ProcGen seems to have 16 tasks. The submission tests their proposed algorithm in only 8 of them.**
>
> Thank you for your suggestion. We presented experimental results for 8 environments because these are the environments we ran. We understand the reviewer's concerns; however, the Procgen benchmark demands significant computational resources (each algorithm requires 50M interactions with the environment and must be run with 3 different random seeds independently, which is much larger than Atari and MuJoCo). Despite this, our framework still achieved substantial improvements in generalization performance (as shown in Figure 5 for bigfish, bossfight, chaser, coinrun, and fruitbot). Additionally, we referenced several accepted high-quality conference papers that also use Procgen as a benchmark, and we found that [1] used only 4 environments, [2] used only 3 environments, and [3] used 7 environments. This suggests that the number of environments should not be a strict criterion, especially considering our significant performance improvements.
>
> >**More recent techniques report substantially higher scores in the ProcGen environment.**
>
> Thank you for your reminder. We want to emphasize that: **our dual-agent method serves as a plug-and-play framework which can be applied to various of RL algorithms, ranther than a newly proposed RL algorithm.** Hence, we do not emphasize a direct performance comparison between our method and other state-of-the-art algorithms. Instead, we highlight the performance improvement when combining our method with PPO, demonstrating that our proposed framework indeed contributes to generalization. However, further investigating the similarities and differences between these methods and ours would be very interesting.
>
> >**Another thing I want to mention is that by employing the proposed adversarial learning framework the number of encoder parameters that needs to be trained is in fact doubled. This brings a new set of questions. Is it really a fair comparison to lower capacity models as previous ones? Would the prior methods perform also well if we simply just increased the parameters in the encoder?**
>
> This is indeed a good question. Choosing an encoder with double the number of parameters for comparison is challenging, although we believe that simply doubling the parameters of the PPO encoder would not match the performance of our method. Therefore, we also presented the training curve of DAAC for reference. While you pointed out that IDDAC seems to perform better, we found that DAAC and IDDAC have similar performance, with DAAC sometimes performing better in certain environments but is much simpler to implement, making it a natural choice. Nevertheless, we apologize for the lack of a broader comparison with other baselines.
>
> ## References
>
> [1] D Ghosh et al. Why generalization in rl is difficult: Epistemic pomdps and implicit partial observability.
>
> [2] M Laskin et al. Reinforcement learning with augmented data.
>
> [3] Z Jia et al. Improving policy optimization with generalist-specialist learning.
>
> [4] R Raileanu et al. Decoupling value and policy for generalization in reinforcement learning.

---

### Official Review · Reviewer_sf2U · 2024-11-04

**Soundness:** 4
**Presentation:** 3
**Contribution:** 4
**Rating:** 5
**Confidence:** 3

**Summary:**

The paper introduces a dual-agent adversarial framework to improve generalization in reinforcement learning (RL). In this setup, two agents interact adversarially, each attempting to disrupt the other's policy while maintaining stability in its own. This competition drives both agents to develop robust and generalizable strategies. The framework is efficient, adding only one hyperparameter, and shows strong performance improvements in challenging environments, especially when used with standard RL algorithms like PPO. This approach offers a promising solution for enhancing RL generalization without relying on complex data augmentations or human-designed biases.

**Strengths:**

1. **Solid Theoretical Background**: The approach is backed by strong theory, clearly explaining how it supports RL generalization.
2. **No Human Bias in Addressing Generalization**: The method achieves generalization without relying on human biases, such as hand-designed augmentations.
3. **No Extra Network Parameters**: The framework achieves its goals without adding network parameters, relying on just one hyperparameter for flexibility.
4. **Novel Idea**: The dual-agent adversarial setup is an innovative way to tackle RL generalization.
5. **Strong Performance**: The approach performs well across tested environments, demonstrating robust generalization and effectiveness.

**Weaknesses:**

1. **Limited Environments and Baselines**: Testing is somewhat limited in environments and baseline comparisons. Adding diverse environments, such as DMC-GB[1], and competitive baselines like PIE-G[2], SVEA[3], and ARPO[4] would provide a more complete comparison and further demonstrate the model's capabilities.

[1] Generalization in reinforcement learning by soft data augmentation., Hansen et al., ICRA 2021

[2] Pre-Trained Image Encoder for Generalizable Visual Reinforcement Learning., Yuan et al., NeurIPS 2022

[3] Stabilizing deep q-learning with convnets and vision transformers under data augmentation., Hansen et al., NeurIPS 2021

[4] Adversarial Style Transfer for Robust Policy Optimization in Deep Reinforcement Learning., Rahman et al., ArXiv 2023

**Questions:**

Please refer to the Weakness

---

> ### Author Response · Authors · 2024-11-28
>
> Dear Reviewer sf2U,
>
> We greatly appreciate your high praise for our work!
>
> >**Limited Environments and Baselines: Testing is somewhat limited in environments and baseline comparisons. Adding diverse environments, such as DMC-GB[1], and competitive baselines like PIE-G[2], SVEA[3], and ARPO[4] would provide a more complete comparison and further demonstrate the model's capabilities.**
>
> Thank you for your suggestions. In this work, we have developed a theoretical framework to guide the generalization of reinforcement learning. However, due to the high computational demands of Procgen, it is challenging to add more baseline algorithms in a short amount of time, and we apologize for this.
>
> We want to emphasize that: **our dual-agent method serves as a plug-and-play framework which can be applied to various of RL algorithms, ranther than a newly proposed RL algorithm.** Hence, we present the significant performance improvement achieved when combining PPO with our framework, rather than comparing performance with other SOTA algorithms. It is worth noting that the main baseline in [4] is also PPO, and we have additionally included the more classical DAAC algorithm [5] as a reference baseline.
>
> ## References
>
> [1] N Hansen et al. Generalization in reinforcement learning by soft data augmentation.
>
> [2] Z Yuan et al. Pre-trained image encoder for generalizable visual reinforcement learning.
>
> [3] N Hansen et al. Stabilizing deep q-learning with convnets and vision transformers under data augmentation.
>
> [4] MM Rahman et al. Adversarial Style Transfer for Robust Policy Optimization in Deep Reinforcement Learning.
>
> [5] R Raileanu et al. Decoupling value and policy for generalization in reinforcement learning.

---

### Official Review · Reviewer_tBzc · 2024-11-04

**Soundness:** 3
**Presentation:** 3
**Contribution:** 3
**Rating:** 6
**Confidence:** 3

**Summary:**

This paper introduces a novel adverserial learning framework that involves a minimax game process between two homogeneous agents to  improve the generalization capability of these agents in RL. This framework integrates with existing RL algorithms such as PPO, leverages no additional human prior knowledge which can lead to poor robustness in generalization and has minimal hyperparameters allowing for effective applicability. The authors additionally derive lower bounds for the training and generalization performance of the agent and show that by minimizing the policy's robustness to irrelevant features, one can improve generalization performance. The authors evaluate their framework in the ProcGen environment, showing gains over algorithms such as PPO and DACC.

**Strengths:**

- Empirically show a significant improvement over prior work in the ProcGen environment with their adverserial learning framwork
- Provide theoretical insights about how a policy's robustness to irrelevant features improves generalization performance which is a novel contribution that can be generally applied to any algorithm.

**Weaknesses:**

- Several works such as DRAC and RARL consider a multi-agent/adversarial optimization process in RL. Would be good to include an extensive evaluation of these approaches as baselines and contextualize the novelty of your approach with respect to each baseline.
- The method is primarily evaluated in the ProcGen environment and could benefit from additional empirical evaluation with a larger set of RL benchmarks to further evaluate the efficacy of the approach.
- GANs and other adversarial optimization techniques commonly have issues with mode collapse, vanishing gradients and convergence issues, which all make optimization more difficult. Though this is controlled with the parameter $\alpha$, would be good to consider the tradeoff of the robustness to adversarial threats and the performance of the agent.

**Questions:**

- One claim is that the method can be widely used with a variety of algorithms. Would you be able to share results on how this approach transfers with different Online RL algorithms in the ProcGen benchmark?
- Would you be able to provide some qualitative analysis of the representations learned by your framework compared with those of PPO and DACC to further validate the claim that robust representations are being learned?
- Could you share results of the sensitivity of $\alpha$ and the selection criterion for it?

---

> ### Author Response · Authors · 2024-11-28
>
> Dear Reviewer tBzc,
>
> Thank you very much for your support of our work! We will now address your concerns.
>
> >**Several works such as DRAC and RARL consider a multi-agent/adversarial optimization process in RL. Would be good to include an extensive evaluation of these approaches as baselines and contextualize the novelty of your approach with respect to each baseline.**
>
> Thank you for your suggestion. We want to emphasize that: **our dual-agent method serves as a plug-and-play framework which can be applied to various of RL algorithms, ranther than a newly proposed RL algorithm.** Hence, we did not include these additional baselines. However, further investigating the similarities and differences between these methods and ours would be very interesting.
>
> >**The method is primarily evaluated in the ProcGen environment and could benefit from additional empirical evaluation with a larger set of RL benchmarks to further evaluate the efficacy of the approach.**
>
> Thank you very much for your suggestion. We note that the Procgen environments are already one of the most widely used benchmarks for testing generalization performance [1, 2, 3, 4, 5], so we primarily focused on testing within Procgen. We greatly appreciate your thoughtful comments.
>
> >**GANs and other adversarial optimization techniques commonly have issues with mode collapse, vanishing gradients and convergence issues, which all make optimization more difficult. Though this is controlled with the parameter $\alpha$, would be good to consider the tradeoff of the robustness to adversarial threats and the performance of the agent.**
>
> >**Could you share results of the sensitivity of $\alpha$ and the selection criterion for it?**
>
> Thanks for your comments. We must admit that the current default value of \(\alpha\) is set empirically to 1. Due to the high computational demands of Procgen, it may be difficult to obtain this analysis result in a short amount of time.
>
> >**One claim is that the method can be widely used with a variety of algorithms. Would you be able to share results on how this approach transfers with different Online RL algorithms in the ProcGen benchmark?**
>
> Thank you for your valuable question. We understand your concerns. We chose PPO as the baseline because it is one of the most widely used algorithms in reinforcement learning, such as in RLHF, and humanoid robots typically use PPO as the default algorithm. Another consideration is that the PPO algorithm outputs a probability distribution over the action space, which aligns with the KL divergence calculation in (19). Although it is challenging to provide results combining other baselines with our method in a short amount of time, we would like to emphasize the theoretical contributions of this paper.
>
> >**Would you be able to provide some qualitative analysis of the representations learned by your framework compared with those of PPO and DACC to further validate the claim that robust representations are being learned?**
>
> We understand your concerns, but we believe that good generalization performance can demonstrate this. You can observe a significant improvement in generalization performance from Figure 5, which is direct evidence that the agent is learning more robust representations.
>
>
>
> ## References
>
> [1] K Cobbe et al. Leveraging procedural generation to benchmark reinforcement learning.
>
> [2] K Cobbe et al. Quantifying generalization in reinforcement learning.
>
> [3] R Raileanu et al. Decoupling value and policy for generalization in reinforcement learning.
>
> [4] R Raileanu et al. Automatic data augmentation for generalization in deep reinforcement learning.
>
> [5] A Jesson et al. Improving Generalization on the ProcGen Benchmark with Simple Architectural Changes and Scale.

---

### Official Review · Reviewer_g3LS · 2024-11-04

**Soundness:** 3
**Presentation:** 3
**Contribution:** 3
**Rating:** 5
**Confidence:** 4

**Summary:**

This paper introduces a dual-agent adversarial policy learning framework to address generalization gaps in reinforcement learning (RL). The authors first derive a lower bound on generalization performance, showing that optimizing this bound corresponds to constrained optimization at each RL step. They then leverage some approximations, leading to the dual-agent adversarial framework proposed in this paper. It employs two identical policy networks that are updated alternately to minimize reliance on irrelevant features through a combined loss function, which includes both the primary task loss and a new adversarial loss that includes both adversarial attacks of the other and robust defences of itself. Experiments on the ProcGen benchmark show that this approach outperforms PPO and DAAC baselines in generalization.

**Strengths:**

The paper is well-written and easy to follow, with an effective and engaging presentation. The paper begins with a theoretical analysis, systematically deriving the motivation and key designs, and ultimately showing good results.

**Weaknesses:**

My main concern is the evaluation. Although the performance is promising, the paper lacks in-depth analysis and extensive discussion on several aspects.

Currently, it seems that even without generalization, the proposed method is showing good performance. Thus it is unclear if it is due to better convergence or better generalization capability. We should be careful about this when drawing the conclusion that the proposed method has better generalization performance. And it would be helpful to add some experiments/discussion to compare only the generalization if the two baselines have similar in-distribution performance. Also, one needs to check if the comparison is fair or not. It would be helpful to report the wall clock time and number of gradient steps between the proposed method and the baseline.

Second, the approach is evaluated on only one generalization setup. However, it is unclear how challenging such a generalization setting is. It would be beneficial to conduct further analysis, assessing the degree to which the proposed algorithm enhances generalization across varying levels of difficulty, such as easy, moderate, and challenging generalization cases.

Third, please also consider adding the training complexity compared to baseline methods, as well as a discussion on the impact of hyperparameter $\alpha$.

Additionally, the authors may want to compare and discuss their method relative to other approaches aimed at enhancing RL generalization, such as [1][2]. Given that these methods may also align with the two characteristics described in Section 4.2, further elaboration on the distinctions or similarities would strengthen the paper.

* [1] MaDi: Learning to Mask Distractions for Generalization in Visual Deep Reinforcement Learning
* [2] Policy Rehearsing: Training Generalizable Policies for Reinforcement Learning

In Section 2, the introduction of the concept of MDP state semantics and the subscript $𝑚$ in the MDP notation is not well-motivated or clearly explained. Consider improving clarity by first defining the distribution $𝑝𝑀$ explicitly and then introducing $m$. Furthermore, the limitations and future works should be discussed in the paper.

**Questions:**

* Will the theorem hold if $𝑀_{train}$ is unbounded? In RL, policies typically interact with the environment on the fly, allowing for the gathering of infinite samples.
* Are the two characteristics discussed in Section 4.2 sufficient to ensure robust generalization performance? If not, what additional considerations could be relevant?
* Are any weights needed in Equation 19? If not, why?
* Would it be beneficial to consider heterogeneous encoders in the proposed method?

---

> ### Author Response · Authors · 2024-11-28
>
> Dear Reviewer g3LS,
>
> Thank you very much for your support of our work. We will address your concerns below.
>
> >**Currently, it seems that even without generalization, the proposed method is showing good performance. Thus it is unclear if it is due to better convergence or better generalization capability. We should be careful about this when drawing the conclusion that the proposed method has better generalization performance.**
>
> This is a good question! However, there is no need to worry, as we have already proven Theorem 3.2 (which only relies on the mild assumptions of 3.1):
> $$\zeta(\pi)\geq\eta(\pi)-\frac{2r_{\max}}{1-\gamma}\cdot(1-M)$$
> which means that when optimizing the training performance $\eta(\pi)$, the lower bound of the generalization performance $\zeta(\pi)$ is also optimized. $M$ represents the integral of the training set $\mathcal{M}_{train}$ over the entire $\mathcal{M}$. Moreover, as $M$ increases, the lower bound on generalization performance is gradually optimized. Figure 2 of paper [1] provides strong evidence for this theoretical result of ours.
>
> >**Second, the approach is evaluated on only one generalization setup. However, it is unclear how challenging such a generalization setting is. It would be beneficial to conduct further analysis, assessing the degree to which the proposed algorithm enhances generalization across varying levels of difficulty, such as easy, moderate, and challenging generalization cases.**
>
> Thank you for your valuable suggestion. However, we would like to clarify that the difficulty of the Procgen environments is typically set to either "easy" or "hard" [1, 2], with the "easy" environments being almost non-challenging, where simply using PPO can effectively train the agent [1]. Therefore, we chose the "hard" difficulty to introduce more challenge, and we observed significant improvements, particularly in the bigfish, chaser, and fruitbot environments as shown in Figure 5.
>
> >**Additionally, the authors may want to compare and discuss their method relative to other approaches aimed at enhancing RL generalization, such as [3, 4].**
>
> We appreciate the two excellent papers you provided! Given the limited rebuttal period, a detailed comparison of their methods with ours is challenging. However, we would be happy to incorporate them into our future work.
>
> >**Consider improving clarity by first defining the distribution $p_{\mathcal{M}}$ explicitly and then introducing $m$. Furthermore, the limitations and future works should be discussed in the paper.**
>
> Thank you for your incredibly thorough review. We introduce $p_{\mathcal{M}}$ first in line 132. Additionally, this paper does not specifically discuss the limitations, as we believe that the main Theorems 3.2 and 3.5 depend only on the mild Assumption 3.1. We are excited to include a more detailed analysis of our method along with others (such as [3] and [4]) in future work.
>
> >**Will the theorem hold if $\mathcal{M}_{train}$ is unbounded? In RL, policies typically interact with the environment on the fly, allowing for the gathering of infinite samples.**
>
> Our theorem still holds whether $p_{\mathcal{M}}$ is bounded or unbounded. If it is bounded, the probability distribution is discrete; otherwise, it is continuous. This does not affect the main conclusions of the paper.

---

> ### Author Response · Authors · 2024-11-28
>
> >**Are the two characteristics discussed in Section 4.2 sufficient to ensure robust generalization performance? If not, what additional considerations could be relevant?**
>
> We believe that these two conditions are necessary for the agent to achieve good generalization performance. There may be other conditions as well, but at a minimum, these two must be satisfied.
>
> >**Are any weights needed in Equation 19? If not, why?**
>
> Thanks for your question. In the current version, no weights are required in (19), as the two KL divergences are of the same order of magnitude.
>
> >**Would it be beneficial to consider heterogeneous encoders in the proposed method?**
>
> Thank you for your insightful suggestion. Within the current framework, homogeneous encoders are not strictly necessary, but since the data from both agents need to be simultaneously input to each other's encoders, using homogeneous encoders appears to be a natural choice. We would be happy to explore this further in future work.
>
> ## References
>
> [1] K Cobbe et al. Leveraging procedural generation to benchmark reinforcement learning.
>
> [2] K Cobbe et al. Quantifying generalization in reinforcement learning.
>
> [3] B Grooten et al. MaDi: Learning to Mask Distractions for Generalization in Visual Deep Reinforcement Learning.
>
> [4] C Jia et al. Policy Rehearsing: Training Generalizable Policies for Reinforcement Learning.

---

> > ### Comment · Reviewer_g3LS · 2024-12-01
> >
> > I thank the authors for the rebuttal and the clarifications provided. However, the response does not effectively address my concerns, as much of the reply consists of promises of future experiments and discussions rather than providing concrete evidence or resolutions. Many rebuttal points are presented as assertions without detailed justification or supporting evidence. For example, significant concerns remain regarding the empirical evaluation of generalization, which seems to be the key contribution of this paper. I thus retain my initial score as 5.

---

> > > ### Author Response · Authors · 2024-12-02
> > >
> > > Dear Reviewer g3LS,
> > >
> > > We fully understand your concerns. Thank you once again for your valuable suggestions on our work.

---

### Author Response · Authors · 2024-11-28
**General Comment**

We thank all the reviewers for thoroughly and carefully reading our paper. Below, we address the primary concerns raised by the reviewers:

**1. How to clarify if the improved results is due to better convergence or better generalization capability?**

Theorem 3.2 theoretically proves that when optimizing the training performance, the lower bound of the generalization performance is also optimized. As the integral of the training set increases, the lower bound on generalization performance is gradually optimized. Hence, the improved performance directly supports the improvement of the generalization capability.


**2. Limited environments and baseline comparisons.**

- Firstly, the Procgen environments are already one of the most widely used benchmarks for testing generalization performance, so we primarily focused on testing within Procgen.
- Secondly, We want to emphasize that: **our dual-agent method serves as a plug-and-play framework which can be applied to various of RL algorithms, ranther than a newly proposed RL algorithm.** Hence, we present the significant performance improvement achieved when combining PPO with our framework, rather than comparing performance with other SOTA algorithms.

**3. Theoretical insights.**

Finally, we would like to gently remind all reviewers not to overlook the theoretical contributions of this work. All the theoretical results in this paper rely solely on the mild Assumption 3.1. In particular, reviewer tBzc pointed out: **Provide theoretical insights about how a policy's robustness to irrelevant features improves generalization performance, which is a novel contribution that can be generally applied to any algorithm.**

We greatly respect the reviewer's opinion on adding experiments, which is of great help to the integrity of the article. However, due to time and resources constraints, we hard to supplement more experiments, we will add more experiments in the future, thank you!


Thanks for your time and effort in reviewing our paper, your suggestions have greatly helped to enhance the article!

---

### Note · Authors · 2025-01-23

I have read and agree with the venue's withdrawal policy on behalf of myself and my co-authors.